# Total genetic contribution assessment across the human genome

Ting Li[1], Zheng Ning [1,2], Zhijian Yang[1], Ranran Zhai [1], Chenqing Zheng[1], Wenzheng Xu[1], Yipeng Wang [1], Kejun Ying [1,3,4], Yiwen Chen [1,5] & Xia Shen [1,2,6✉]

Quantifying the overall magnitude of every single locus' genetic effect on the widely measured human phenome is of great challenge. We introduce a unified modelling technique that can consistently provide a total genetic contribution assessment (TGCA) of a gene or genetic variant without thresholding genetic association signals. Genome-wide TGCA in five UK Biobank phenotype domains highlights loci such as the *HLA* locus for medical conditions, the bone mineral density locus *WNT16* for physical measures, and the skin tanning locus *MC1R* and smoking behaviour locus *CHRNA3* for lifestyle. Tissue-specificity investigation reveals several tissues associated with total genetic contributions, including the brain tissues for mental health. Such associations are driven by tissue-specific gene expressions, which share genetic basis with the total genetic contributions. TGCA can provide a genome-wide atlas for the overall genetic contributions in each particular domain of human complex traits.

[1] Biostatistics Group, State Key Laboratory of Biocontrol, School of Life Sciences, Sun Yat-sen University, Guangzhou, China. [2] Department of Medical Epidemiology and Biostatistics, Karolinska Institutet, Stockholm, Sweden. [3] Division of Genetics, Department of Medicine, Brigham and Women's Hospital and Harvard Medical School, Boston, MA, USA. [4] Harvard T.H. Chan School of Public Health, Harvard University, Boston, MA, USA. [5] Department of Cell and Molecular Biology, Uppsala University, Uppsala, Sweden. [6] Centre for Global Health Research, Usher Institute, University of Edinburgh, Edinburgh, UK. ✉email: xia.shen@ed.ac.uk

Understanding the magnitude of ubiquitous genetic effects across the genome is of fundamental importance for scrutinising the complexity of genetic architecture. The concept of pleiotropy, coined 110 years ago by the German geneticist Ludwig Plate[1], describes the phenomenon of a single genetic variant or gene affecting multiple phenotypes, reflecting the shared genetic basis across these traits[2]. During the past decade, a large amount of genome-wide association studies (GWAS) of various phenotypes have shown that complex traits are highly polygenic, influenced by many genetic variants with small effects[3]. When many phenotypes are considered, distinguishing tiny genetic effects from noise becomes particularly challenging. Not only the spread of genetic effects, i.e. pleiotropy, is of concern, but also the magnitude of genetic effects requires consideration. Which variant, or gene, or part of the genome, contributes the most genetic effects to the measured human phenome? The question is yet to be well defined and scientifically answered.

Quantifying the overall effect of a gene or genetic variant on the human phenome is not trivial. Although this seems to be as simple as adding up each variant's genetic effect on each phenotype, there are some major difficulties. The key issue is that it is hard to distinguish associated phenotypes from noise. Simply applying a statistical significance threshold to association summary statistics cannot solve the problem[4]. For instance, in GWAS setting, a loose threshold (e.g. $P < 0.05$) may lead to too many false positives, while a stringent threshold (e.g. $P < 5 \times 10^{-8}$) may produce too many false negatives[5]. Without a significance threshold, summing up the association statistics across all phenotypes also suffer from the issue of too many negatives, so that noise dominates signals. Thus, it is strongly needed to develop a threshold-free method, to penalise the unassociated phenotypes and quantify the total contribution of a gene or genetic variant[6]. Correlations across a wide range of phenotypes also need to be accounted for[4].

In this work, we develop a statistic for total genetic contribution assessment (TGCA) of a genetic variant, addressing the above difficulties. The TGCA statistic quantifies how much genetic effect in total a SNP contributes to a group of phenotypes. We achieve this by modeling GWAS summary statistics across a wide range of phenotypes in UK Biobank (UKBB). We integrate TGCA with tissue-specific gene-expression information to highlight the tissues most relevant and irrelevant to human complex traits. We also provide functional insights into the discoveries by integrating TGCA with functional genomic annotations. TGCA results can be considered as an atlas of general genetic architecture for complex trait variation.

## Results

**Overview of the methods**. We introduce a threshold-free method assessing the total genetic contribution for each single SNP based on GWAS summary statistics of a series of phenotypes. Prior to statistical modeling, we standardise the GWAS Z-score test statistics across different phenotypes by the square root of the sample sizes, so that the subsequent modeling would not be biased due to unequal study-size-driven power. Subsequently, we adjust the phenotypic correlations among the Z-scores[4], for every single genetic variant across multiple phenotypes (see Methods), yielding a vector of adjusted Z-scores per variant that are uncorrelated with each other. Denote the adjusted Z-scores of SNP $i$ across $k$ phenotypes as $\mathbf{z}_i = (z_{i1}, \ldots, z_{ik})$. In the TGCA model, for $j \in \{1, \ldots, k\}$, $z_{ij}$ is drawn from a mixture Gaussian distribution of

$$\pi_- N(\mu_-, \sigma_1^2) + \pi_0 N(0, 1) + \pi_+ N(\mu_+, \sigma_2^2), \qquad (1)$$

where the three components correspond to the negative effects, the null effects, and the positive effects. The proportions of the components are constrained by $\pi_- + \pi_0 + \pi_+ = 1$. The $\mu$ parameters are the means of the genetic effects, and the $\sigma^2$ parameters measure the dispersion of the genetic effects. We estimate all these parameters for each genetic variant in the genome by maximising the full likelihood via an EM algorithm (Supplementary Information).

In order to quantify the total genetic effect per variant, we define a TGCA statistic $\Theta = |\pi_+ \mu_+| + |\pi_- \mu_-|$. Different from an ordinary pleiotropy concept, which one might define as a SNP affecting more than one trait, the $\Theta$ parameter aims to quantify the total genetic contribution of each SNP on a group of phenotypes, standardised by the number of traits modelled due to the constrain on the $\pi$ parameters. Its main advantage is being threshold-free. The statistic is constructed via the non-null parameters in the above mixture model, so that the $\pi_0 N(0, 1)$ component serves as a penalisation term for phenotypes with null associations without introducing any significance threshold for GWAS summary statistics. $\Theta$ quantifies how much genetic effect in total a SNP contributes to the group of traits, the $\pi$ parameters assess the proportion of these traits the SNP affects, and the $\mu$ parameters represent the average effect the SNP has on each trait.

The $\Theta$ score itself is comparable across SNPs. When inference on the uncertainty and model fitting is needed, standard errors of the $\pi$ and $\mu$ parameters are derived from the Hessian of the likelihood function at its maximum, so that the standard error of estimated $\Theta$ can be obtained via the Delta method (Supplementary Information). A $p$-value testing against the null hypothesis of $\Theta = 0$ can be obtained via a Wald test, i.e. a quadratic approximation of the likelihood around its maximum.

**Simulations**. Although the modeling is straightforward, the identifiability of mixture models is of numerical concern. In order to assess the validity of the TGCA statistic $\Theta$, we simulated different scenarios of models and parameter values. First, for a genetic variant, we generated 200 GWAS Z-scores, corresponding to 200 independent phenotypes, from four different true models: a mixture distribution with two non-null Gaussian components $\pi_- N(\mu_-, \sigma_1^2) + \pi_0 N(0, 1) + \pi_+ N(\mu_+, \sigma_2^2)$, where the $\pi_- = \pi_+$; a mixture with one non-null Gaussian $\pi_0 N(0, 1) + \pi_+ N(\mu_+, \sigma^2)$; a mixture with three non-null Gaussian $\pi_- N(\mu_-, \sigma_1^2) + \pi_0 N(0, 1) + \pi_+ N(\mu_+, \sigma_2^2) + \pi_{++} N(\mu_{++}, \sigma_3^2)$, where $\pi_- = 2\pi_+ = 2\pi_{++}$; and a mixture with two non-null $t$-distributed heavy-tailed components $\pi_- [t(1) + \mu_-] + \pi_0 N(0, 1) + \pi_+ [t(1) + \mu_+]$, where $t(1)$ denotes a $t$-distribution with 1 degree of freedom, and $\pi_- = \pi_+$. In order to cover a wide range of values in the parameter space, the true parameters were drawn randomly from given distributions (Methods). We evaluated whether $\Theta$ and the other parameters can be consistently estimated with different true values setup (Fig. 1, Supplementary Fig. 1). Across the Gaussian mixture models, regardless of the violation of the TGCA model, $\Theta$ could be consistently estimated. For the mixture with extremely heavy-tailed non-null components, although the individual parameters in the mixture were hard to identify, $\Theta$ could still be consistently estimated.

Second, for the scenario where the phenotypes are correlated, we generated correlated Z-scores according to the estimated phenotypic correlations across 1376 traits in the UK Biobank (UKBB)[7]. We then decorrelated the Z-scores using the top 955 eigenvectors corresponding to 90% information of the phenotypic correlation matrix (See Methods). For each of the above Gaussian mixture true models, we estimated the model parameters and compared the estimates to the true values (Supplementary Fig. 2). We found that the individual parameters in the mixture model

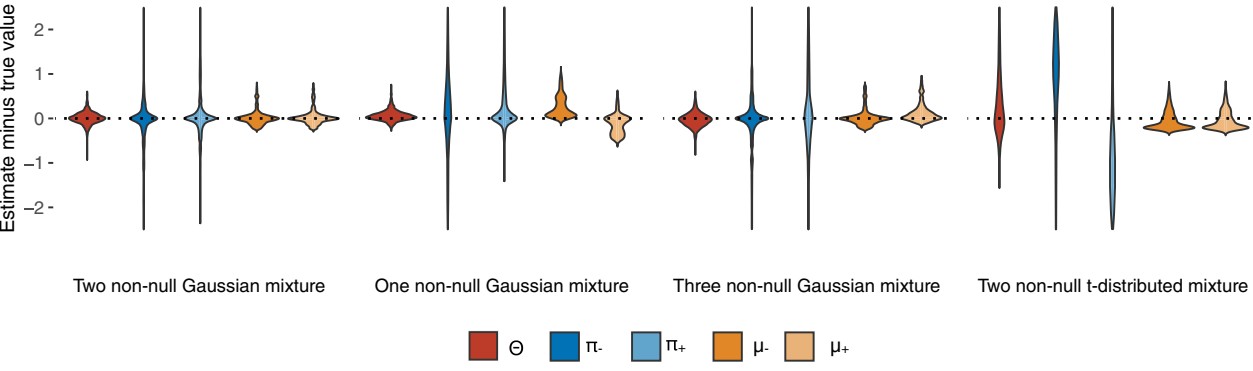

**Fig. 1 Representative simulation results of total genetic contribution assessment (TGCA) under different true models.** 200 independent Z-scores for a single genetic variant were drawn from a mixture distribution with (1) two non-null Gaussian components $0.25N(\mu_-, \sigma_1^2) + 0.5N(0, 1) + 0.25N(\mu_+, \sigma_2^2)$, (2) one non-null Gaussian component $0.5N(0, 1) + 0.5N(\mu_+, \sigma^2)$, (3) three non-null Gaussian components $0.25N(\mu_-, \sigma_1^2) + 0.5N(0, 1) + 0.125N(\mu_+, \sigma_2^2) + 0.125N(\mu_{++}, \sigma_3^2)$, and (4) two non-null heavy-tailed $t$-distributed components $0.25[t(1) + \mu_-] + 0.5N(0, 1) + 0.25[t(1) + \mu_+]$, respectively. The simulation was repeated for 999 times. In each simulation, the negative effect size $\mu_-$ was randomly drawn from $-|N(1, 1)|$, the positive effect size(s) $\mu_+$ and $\mu_{++}$ were drawn from $|N(1, 1)|$, and the $\sigma^2$ parameters from $\chi^2(1)$. The y-axis compares the estimated $\hat{\Theta} = |\hat{\pi}_+ \hat{\mu}_+| + |\hat{\pi}_- \hat{\mu}_-|$ and the relevant parameters with the true values. Source data are provided as a Source Data file.

became harder to identify consistently, especially when some parameters were close to the boundary. Nevertheless, $\Theta$ was identifiable and can be consistently estimated even when the true model violates the TGCA model.

For the four simulated model and a null model with no genetic effect, we evaluated the operating characteristics of the p-values testing the null hypothesis of $\Theta = 0$ (Supplementary Fig. 3). The three models that violated the TGCA model did not substantially affect the performance. The $p$-value distribution under the null, although slightly inflated, was close to uniform. For the same $\Theta$ value, the more the null effect traits were, the higher the power to identify $\Theta$, as the model became more identifiable when the components in the mixture were more apart from each other.

**Genome-wide TGCA across UKBB phenotypes**. In real data application, we first applied the mixture model to the GWAS summary statistics of 1376 UKBB phenotypes and estimated and tested genome-wide TGCA $\Theta$ of 2,029,920 quality-controlled SNPs with minor allele frequencies (MAFs) > 0.005. We obtained all the estimated parameters as well as the TGCA $\Theta$'s with their standard errors (Supplementary Fig. 4). Based on the analysis of such a large number of traits, we compared the established statistics in pleiotropy measurement literature with the estimated $\Theta$'s across the genome. As we introduced, literature assesses the level of pleiotropy at each SNP by thresholding GWAS p-values, e.g. counting associated phenotypes (HOPS $P_n$ statistic[4]) or phenotype domains[2]. The HOPS $P_m$ statistic sums up GWAS $\chi^2$ statistics, trying to assess a total genetic effect, however, without penalising the null associations, the performance is similar to pleiotropy quantification with thresholding (Supplementary Figs. 5–9). TGCA was specifically developed for total effect quantification without statistical cut-off, yielding a new identifiable parameter $\Theta$. Although pleiotropy affects $\Theta$, $\Theta$ quantifies the total genetic contribution rather than the level of pleiotropy; therefore $\Theta$ captured different features in the genome-wide association data (Supplementary Figs. 5–9).

Next, we performed TGCA in five different phenotype domains, including medical conditions (122 traits), mental health (189 traits), physical measures (117 traits), lifestyle (172 traits), and diet (139 traits) (Fig. 2, Supplementary Figs. 10–12). As $\Theta$ by definition is bounded by zero, we mainly focused the results on the estimated $\Theta$ as scores, rather than treating the analysis as a

GWAS with emphasis on the significance of estimated genetic effects. About 95% of the SNPs had estimated $\Theta$ less than 1. For individual loci, we report those that had $\hat{\Theta} > 2$ (Supplementary Data 1–3). For instance, the *HLA* locus had high $\Theta$ estimates for both medical conditions and physical measures. The bone mineral density locus *WNT16* had the highest $\Theta$ estimate for physical measures. The skin tanning locus *MC1R*, as well as the smoking behaviour locus *CHRNA3*, also known to be associated with human lifespan[8], had high $\Theta$ estimates for lifestyle phenotypes.

In the Supplementary Information, we provide further results on the relationship between these genome-wide TGCA and different aspects, including LD levels, numbers of gene–gene interactions and pathways, and genomic functional annotations.

**TGCA suggests tissues associated with certain phenotype domains**. Integration of genome-wide summary association statistics and tissue- or cell-type-level gene-expression data was shown to be able to locate gene-expression-mediated genetic regulation of complex diseases, e.g. schizophrenia[9] and Parkinson's disease[10], to specific tissues. The strategy consists of two steps: (1) Scoring tissue-specific gene expressions to identify specifically expressed genes in each tissue; (2) Estimating and testing the GWAS-analysed trait's heritability enrichment on SNPs annotated on tissue-specifically expressed genes.

First, we adopted the same procedure as described by Skene et al.[9], using stratified LD score regression (S-LDSC)[11] to test the enrichment of total genetic contributions in different tissues. The expression tissue-specificity score of each gene was calculated as the proportion of expression across 48 GTEx tissues (v7). For each tissue, the top 10% tissue-specifically expressed genes were selected, and SNPs on these genes were annotated and passed onto S-LDSC for genome partitioning (Fig. 3a, Supplementary Data 4). The analysis suggested that only for mental health traits, the estimated $\hat{\Theta}$ were substantially enriched in brain tissues. Although the test statistic $Z_{\Theta} = \hat{\Theta}/s.e.(\hat{\Theta})$ does not follow a normal distribution, the correlation structure of $\mathbf{Z}_{\Theta}$ is a sufficient statistic for such enrichment inference[12]. We also observed a monotonic relationship between $\hat{\Theta}$ and LD correlations (Supplementary Fig. 13), which partly justified the use of S-LDSC for genome partitioning and enrichment test.

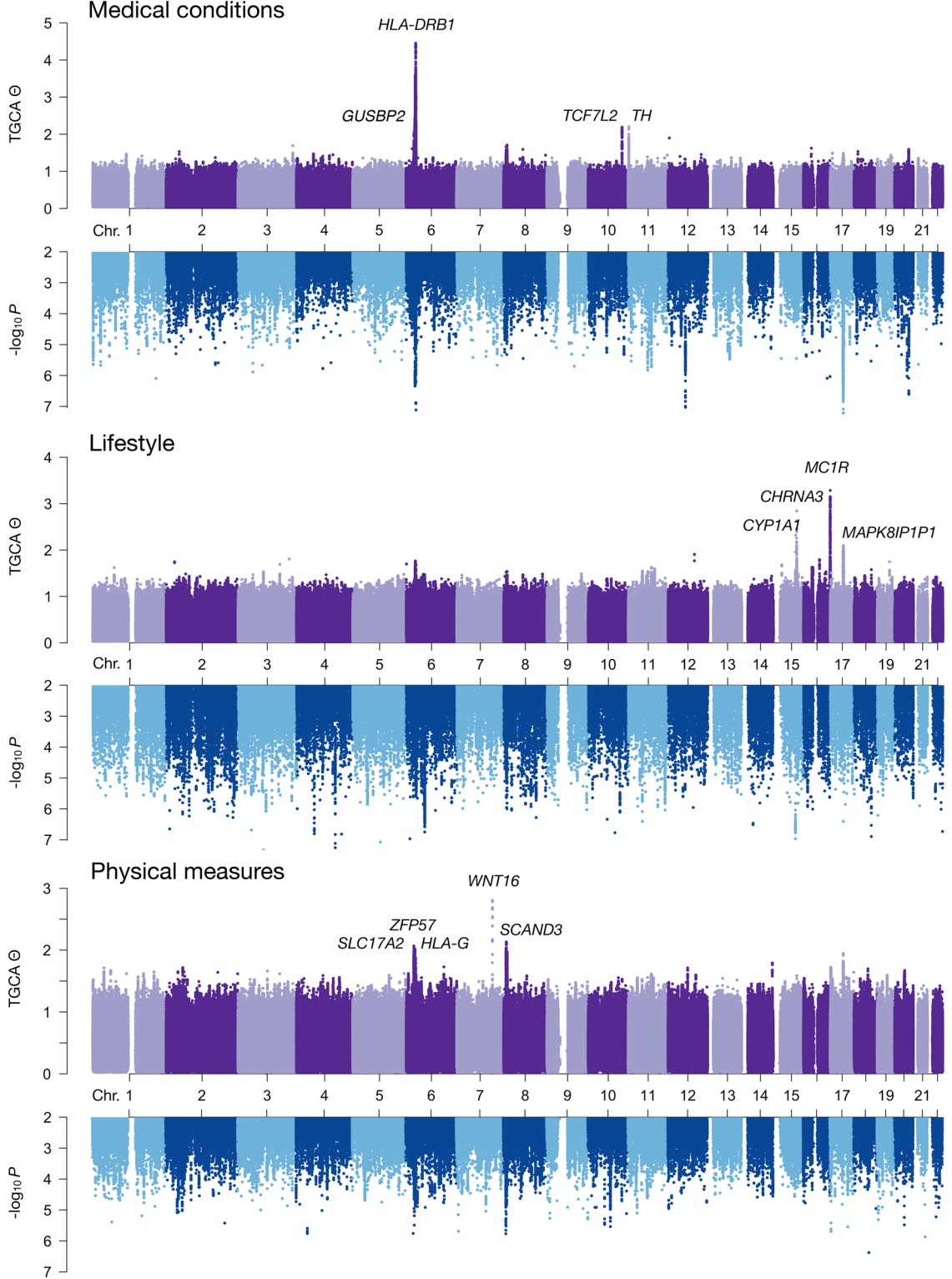

**Fig. 2 Genome-wide total genetic contribution assessment (TGCA) $\hat{\Theta}$ and the $p$-values testing against $\Theta = 0$ in three UK Biobank trait domains.**
Nearest gene(s) to the lead variant at each top locus are labeled. Source data are provided as genome-wide summary statistics in Data Availability.

Nevertheless, secondly, in order to ensure that the majority of the observed enrichment results was not caused by the non-normal distribution of $\hat{\Theta}$, we adopted another strategy similar to the LD-corrected scores in the HOPS method[4]. We regressed the $\hat{\Theta}$ values on each binary annotation variable, defined for each

tissue based on its specifically expressed genes, and the LD scores of the SNPs were used as a covariate. The SNPs included in the regression was LD-pruned to prevent potential bias (see Methods). This alternative analysis strategy also revealed some similar results as S-LDSC, e.g. the estimated $\hat{\Theta}$ for mental health

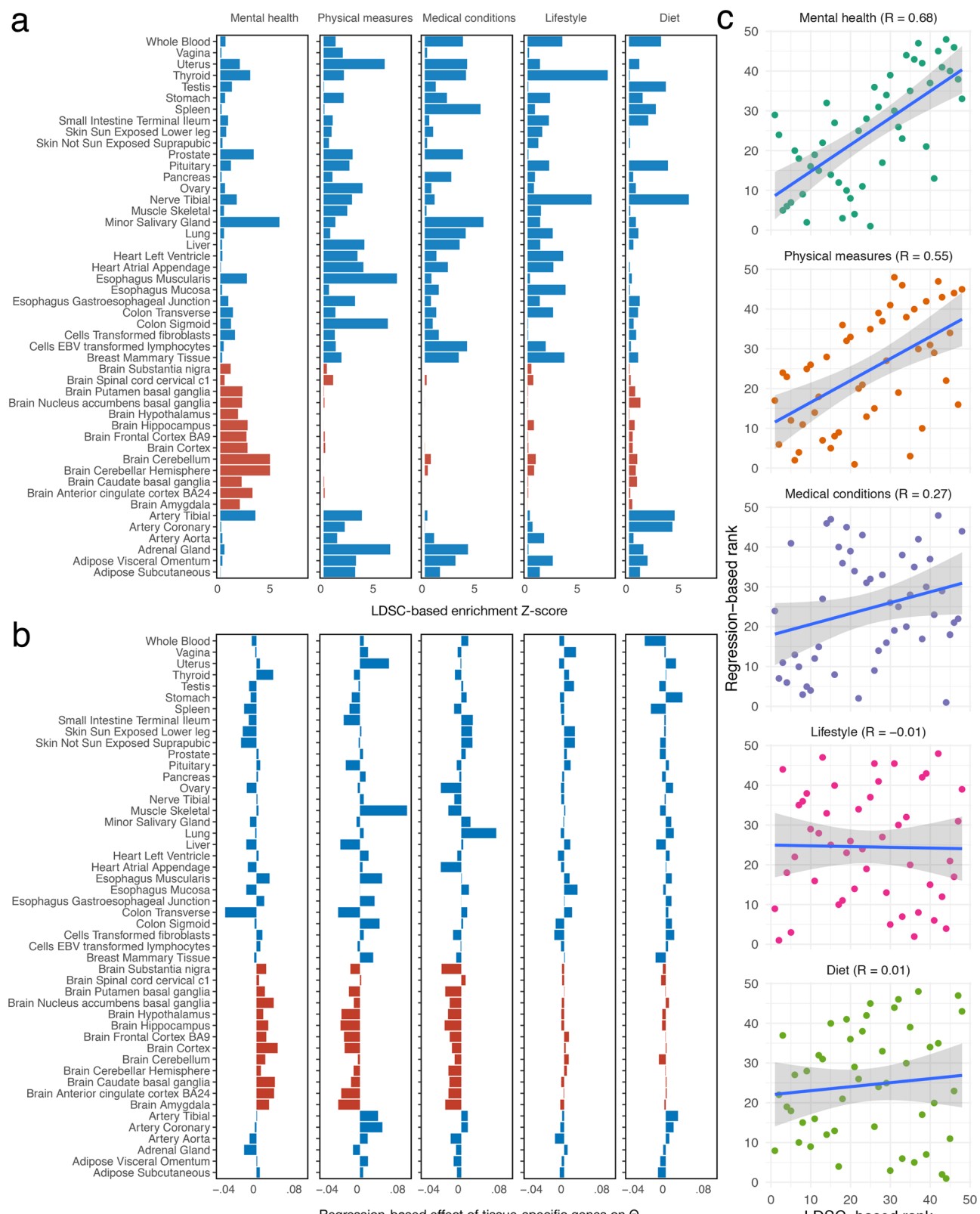

**Fig. 3 Gene-expression-induced association analysis between 48 human tissues and total genetic contribution assessment (TGCA) of five trait domains. a** Association between TGCA Θ and each tissue was tested using stratified LD score regression (LDSC) for the enrichment of Θ in tissue-specifically expressed genes. **b** The Θ-tissue association was analysed using a linear regression of Θ̂ on the annotations of top 10% tissue-specifically expressed genes, corrected for the LD scores of the SNPs. The median regression coefficients of the annotation variables across 100 sets of LD-pruned SNPs are plotted. **c** Comparison of the resulted association scores by the two methods in **a** and **b** via rank-based correlations. The regression lines with the 95% prediction interval bands are shown. Source data are provided as a Source Data file.

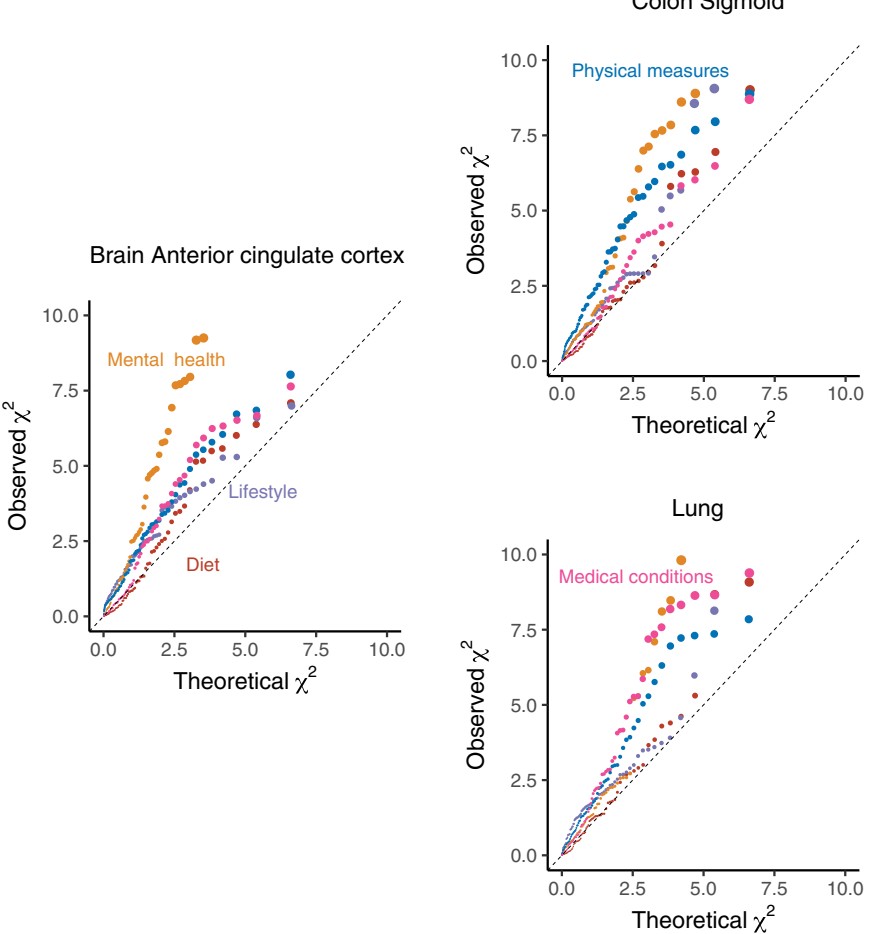

**Fig. 4 Distribution of total genetic contribution assessment (TGCA) $\chi^2$ statistic at *cis*-eQTL of specifically expressed genes in three different tissues.** The quantile-quantile plots compare the observed TGCA $\chi^2$ statistics for $\hat{\Theta}$ at the *cis*-eQTL of top 100 specifically expressed genes in each tissue. The expected null values were drawn from a $\chi^2(1)$ distribution. Source data are provided as a Source Data file.

traits were enriched in brain tissues (Fig. 3b, Supplementary Data 5). As a triangulation strategy, if the results were more similar between the two analysis pipelines for a certain phenotype domain, the tissue enrichment conclusion would be more trustworthy. Thus, we evaluated the rank-based correlation across tissues between the two methods in each phenotype domain (Fig. 3c). The results for mental health, medical conditions, and physical measures showed agreement between the two analysis methods, whereas poor correlations were observed for lifestyle and diet phenotypes. The different correlations reflected different power in detecting the association between $\Theta$ scores and tissue-specific genes. Therefore, for instance, top enriched tissues such as brain for mental health, lung for medical conditions, and sigmoid colon for physical measures, etc., were evident based on these analyses.

**Connection between TGCA and *cis*-regulatory loci of gene expressions.** The *cis*-regulatory variants for gene and protein expressions are centred around transcription start sites (TSS)[13], which explain substantially enriched heritabilities for a wide range of complex traits[11]. With the assessment of total genetic contribution in different trait domains, we expected that the overall genetic contribution would also enrich at *cis*-expression quantitative trait loci (eQTL). Furthermore, if a particular tissue is associated with a certain domain of phenotypes via tissue-specific

gene expressions, we hypothesised that such an association were driven by the *cis*-eQTL of the tissue-specifically expressed genes.

In order to test this, we extracted the *cis*-eQTL scan summary statistics of the tissue-specifically expressed genes in each tissue. The tissue-specifically expressed genes were the same as in the above S-LDSC analysis. We represent each *cis*-eQTL with its lead variant, i.e. the SNP with the smallest *p*-value across the tested *cis*-SNPs by both the eQTL scan and our TGCA analysis. We examined the squared Z-score ($\chi^2$) distribution for $\hat{\Theta}$ in each phenotype domain, stratified on the *cis*-eQTL for specifically expressed genes in each tissue (Fig. 4, Supplementary Fig. 14). As described above, such an investigation might not be trustworthy for lifestyle and diet phenotypes, while evidence could be found for the other phenotype domains. For example, in brain anterior cingulate cortex, the TGCA signals for mental health traits were strong across the eQTL for the particular tissue-specifically expressed genes. Strong TGCA signals for medical conditions were observed at the eQTL for lung-specifically expressed genes. Such signals were also enriched for physical measures in colon sigmoid.

## Discussion
We developed a mixture modelling technique and derived a unified statistic to assess the total genetic contribution of each genetic variant across a wide range of phenotypes. Being

threshold-free and penalising null associations, the TGCA $\Theta$ statistic quantifies an overall genetic effect or a total genetic contribution of a single variant on a group of phenotypes. We applied the method to the UKBB genome-wide summary association statistics for five different domains of phenotypes, highlighted top loci for some phenotype domains, and investigated the genetic regulation mediated via tissue-specific gene expressions. The inferred genome-wide total genetic contribution atlas is useful as a reference, describing the contribution architecture across the human genome. The model itself can be applied to various domains or clusters of phenotypes to assess an overall genetic architecture.

Although the same type of mixture model was not formally evaluated before, we introduced a similar model to assess the proportion of phenotypes that a deletion had effects on[14]. Here we extended from such a modelling technique to develop an identifiable threshold-free score for overall genetic effects. Mixture modelling is flexible, and we expect future development along this line to address relevant scientific questions.

In our model setup, a SNP with high $\pi_+$ and low $\mu_+$ could have the same $\Theta$ as a SNP with low $\pi_+$ and high $\mu_+$. This can apply to both the estimated $\Theta$ and the true $\Theta$, so the model-estimated $\Theta$ itself cannot distinguish the two types of SNPs. The estimated $\pi$ and $\mu$ parameters, although sometimes hard to be consistently identified, can help distinguish these two different scenarios. For the same true value of $\Theta$, the power to identify $\Theta$ varies across different $\pi$–$\mu$ combinations. For the same value of true $\Theta$, the power is higher for low $\pi_+$ and high $\mu_+$ as the locations of the mixture components are more apart and identifiable.

We do not necessarily expect that the effects truly follow the described TGCA model. In fact, we do not know the true model. Nevertheless, there are two advantages of the three-Gaussian TGCA: (1) The parameters are straightforward to interpret, such as on what proportion of the traits the SNP affects (the $\pi$'s), and what the average effect the SNP has on each trait (the $\mu$'s), and how much genetic effect in total a SNP contributes to the traits ($\Theta$); (2) The Gaussian mixture is easier to handle computationally in an EM algorithm. Numerically, in the UKBB analysis, there were about 5% SNPs for which the mixture model fitting could not be stably achieved due to likelihood boundary properties. Some of such lost SNPs could harbour interesting loci, so mathematical and computational algorithms for mixture models might be developed further to increase parameter identifiability.

We found that the number of traits considered in the model does affect the power (Supplementary Fig. 15), nevertheless, for the same true $\Theta$ value, as discussed above, $\pi$–$\mu$ combinations appear to affect the power more. In general, a larger number of traits would increase the power of the mixture model to identify the parameters. Less polygenic traits, e.g. omics phenotypes, can have major loci with relatively larger effects, which may require less number of phenotypes to identify the mixture components.

In order to balance the sample size difference across the traits, we standardised the GWAS Z-scores by the square root of the sample sizes as weights, with a constrain that the average weight across the traits equals to 1. Although modelling the null component variance differently for traits with different sample sizes would fit the data better, modeling the null component as $N(0, 1)$ is reasonable, as the expected variance of the GWAS Z-scores is 1 under the null due to the constrain. We have shown via simulation that the simple model is generally applicable and identifiable in various scenarios.

Given the widespread pleiotropy in the human genome, the power for revealing previously known pleiotropic loci seems to be low if we only look at the $p$-values testing the total genetic contribution parameter against $\Theta = 0$. First, the power for the inference of $\Theta$ and any other parameter in the mixture model increase

with both the sample sizes of the GWASs and the number of phenotypes modelled (Supplementary Figs. 15 and 16). Second, $\Theta$ is a quantity with its own meaning, and the hypothesis testing against $\Theta = 0$ is not very informative. We emphasise the value of $\Theta$ itself rather than how significant it is compared to zero, as the relative magnitudes of $\Theta$'s across different SNPs are more interesting. In order to highlight the top-ranked loci that have not only widespread pleiotropic but also relatively larger effects on many traits, we report the loci with $\hat{\Theta} > 2$, corresponding to the 99.97% quantile for all our $\Theta$ estimates. Nevertheless, the cut-off is arbitrary depending on the aim of the analysis. The threshold-free parameter $\Theta$, regardless of power, provides a unified evaluation across the genome. Similar to GWAS summary statistics of a phenotype, although only a small proportion of loci can be discovered in the GWAS, the whole-genome profile has its own value.

On the technical side, we estimated phenotypic correlations using a simple method based on low-MAF variants, which for the traits analysed, gives estimates very close to the bivariate LDSC intercept estimates (Supplementary Fig. 19). However, the latter requires more computational time and is prone to a loss of estimation efficiency[7], thus we do not recommend the use of bivariate LDSC intercept when estimating phenotypic correlations across a large number of traits. With the estimated phenotypic correlation matrix, the decorrelation step was needed so that the inference of the mixture model became straightforward. Although there are non-unique solutions of the decorrelated $\mathbf{Z}^*$ matrix, as long as it's a linear transformation of $\mathbf{Z}$, the procedure redistributes the estimated genetic effects onto uncorrelated vectors. Therefore, each single value in the $\mathbf{Z}^*$ matrix does not represent an association statistic on an original phenotype anymore, but rather an "association statistic" for a linear combination of many original phenotypes. This is also why, in the context and aim of this work, trying to identify a total genetic contribution parameter $\Theta$ would be more meaningful, rather than identifying which phenotypes the SNP affects. It should be noted that for a large number of traits or an ill-conditioned phenotypic correlation matrix, regularisation of the phenotypic correlation matrix is needed so that the noise gets penalised (Supplementary Fig. 18). Eigen-decomposition enables such regularisation by taking the top eigenvectors. Cholesky decomposition can also achieve the decorrelation purpose (Supplementary Fig. 17), but the algorithm could be problematic for ill-conditioned correlation matrix and not suitable for matrix regularisation.

## Methods

**Summary association statistics in UK Biobank**. We downloaded the UK Biobank round 2 genome-wide association study (GWAS) summary statistics released by the Neale's lab. The GWAS was conducted on 361,194 genomic British individuals (194,174 females and 167,020 males) for 4203 phenotypes. The analysis corrected from baseline characteristics including sex, age, age², age × sex, age² × sex, and 20 genomic principal components. UK Biobank has full ethical approval from the NHS National Research Ethics Service (11/NW/0382; 16/NW/0274).

**Quality control of the phenotypes**. We selected the phenotypes that were harmonised via PHESANT transformation[15]. For continuous and ordinal phenotypes, we filtered out the phenotypes for which there were less than 50,000 non-missing samples. For binary phenotypes, we filtered out those with less than 1000 cases. This resulted in 1511 phenotypes in total, including 1036 binary, 237 continuous, and 238 ordinal phenotypes (Supplementary Data 6).

**Estimation of phenotypic correlations**. In order to model independent test statistics from multiple GWAS, the correlation between the genetic effects of each pair of phenotypes, $\beta_1$ and $\beta_2$, needs to be considered. As the genetic effect per single nucleotide polymorphisms (SNP) $\beta_{ij}$ is tiny, for each SNP $j$, the correlation between two estimates $\hat{\beta}_{1j}$ and $\hat{\beta}_{2j}$ is essentially the correlation between the residuals in the GWAS linear regression model, which is approximately the correlation between the two phenotypes.

Under the null hypothesis of $\beta_1 = \beta_2 = 0$, i.e. no genetic effect, the Pearson's correlation coefficient between the two estimated genetic effects vectors $\hat\beta_1$ and $\hat\beta_2$ provides an unbiased estimator of their phenotypic correlation[16]. Such an estimator has two major advantages: (1) it naturally incorporates the level of sample overlap between the two GWAS, which reduces to zero when the two GWAS are performed on independent samples; (2) it directly produces phenotypic correlations for binary phenotypes on corresponding continuous logistic scale, making the estimates mathematically more straightforward to handle. In practice, it is tricky to select SNPs with zero true genetic effects. The above estimation procedure is valid as long as the genetic variance captured by each selected SNP is small. Thus, instead of putting constraints on estimated SNP effects, we selected the 169,537 SNPs that have minor allele frequency (MAF) less than $5 \times 10^{-4}$ and used the correlation matrix of their test statistics (Z-scores) as the estimate of the phenotypic correlation between the 1511 traits, which is also an estimate of the correlation matrix of the GWAS test statistics for each single variant in the genome. The use of such low-MAF variants provides more efficient estimates of the phenotypic correlations than the other state-of-the-art methods[7]. We removed the phenotypes that have an estimated correlation larger than 0.9 with another trait. This resulted in an estimated phenotypic test statistic correlation matrix $\mathbf{R}$ for 1376 traits (Supplementary Data 7).

**Quality control of the SNPs.** We selected 10,301,794 SNPs, whose MAFs were greater than 0.005 to be the candidate evaluated SNPs. For each candidate evaluated SNP, Kolmogorov–Smirnov test was used to test whether its p-values across GWAS of 1376 filtered traits deviated from the uniform distribution. We selected 2,787,891 SNPs with $p$-values less than 0.1 as the final set of SNPs to be evaluated.

**Adjusting phenotypic correlations.** Denote the Z-score matrix of the $k$ phenotypes on the columns and 2,787,891 SNPs on the rows as $\mathbf{Z}$, re-weighted by the square roots of the sample sizes $\mathbf{N} = (N_1, \dots, N_k)$, i.e. $z_{ij} = w_j \hat\beta_{ij} / \sqrt{\operatorname{var}(\hat\beta)}$, where $w_j = \frac{1}{k} \sum_{i=1}^{k} \sqrt{N_i} / \sqrt{N_j}$. $\mathbf{R}$ is the correlation matrix of the columns of $\mathbf{Z}$. We have $\mathbf{Z}'\mathbf{Z} = c\mathbf{R}$, where $c$ is a constant. As $\mathbf{R}$ is symmetric and semi-positive definite, $\exists \mathbf{Z}^* = \mathbf{Z}\mathbf{R}^{-1/2}$ so that $\mathbf{Z}^{*'}\mathbf{Z}^* = \mathbf{R}^{-1/2}\mathbf{Z}'\mathbf{Z}\mathbf{R}^{-1/2} = c\mathbf{R}^{-1/2}\mathbf{R}\mathbf{R}^{-1/2} = c\mathbf{I}$, i.e. the adjusted test statistics as the columns of $\mathbf{Z}^*$ are uncorrelated, which is similar to the decorrelation procedure in the HOPS method[4]. We obtained $\mathbf{R}^{-1/2}$ via eigen-decomposition of the estimated $\mathbf{R}$ matrix. We approximated the $\mathbf{R}^{-1/2}$ matrix by its first $M$ eigenvectors, yielding $\mathbf{Z}^* = \mathbf{Z}\mathbf{R}^{-1/2}$ with $M$ columns, where $M$ is determined with a cut-off on the eigenvalues so that the eigenvectors capture 90% of the information in $\mathbf{R}$. For each SNP, we conduct modeling of the distribution of the $\mathbf{Z}^*$ values.

**Phenotype domains.** Among the 1,376 QCed phenotypes, 122 traits were classified into the medical conditions domain, 189 traits were classified as mental health traits, 117 traits as physical measures, 172 traits as lifestyle traits, and 139 traits were classified as diet-related according to the UK Biobank phenotype catalogue (Supplementary Data 8–12).

**Statistical modelling of TGCA.** For each SNP $j$, the adjusted GWAS test statistics $\mathbf{Z}_j^*$ were modelled as drawn from a mixture Gaussian distribution of $\pi_- N(\mu_-, \sigma_1^2) + \pi_0 N(0,1) + \pi_+ N(\mu_+, \sigma_2^2)$, where the $\pi$ parameters represent the proportions of negative effects, null effects, and positive effects, and $\pi_- + \pi_0 + \pi_+ = 1$. The $\mu$ parameters denote the means of the genetic effects, and $\sigma^2$ parameters measure the effects dispersions. By maximizing the full likelihood via an EM algorithm, implemented in the R/mixtools package, we estimated all parameters in the model. We define $\Theta = |\pi_+ \mu_+| + |\pi_- \mu_-|$ as a unified quantification of the total genetic contribution per SNP.

**Simulations with independent Z-scores.** We generated 200 independent Z-scores for a single genetic variant that came from four scenarios of true models: Two non-null Gaussian mixture distribution $\pi_- N(\mu_-, \sigma_1^2) + \pi_0 N(0,1) + \pi_+ N(\mu_+, \sigma_2^2)$, where the $\pi_- = \pi_+$; One non-null Gaussian mixture distribution $\pi_0 N(0,1) + \pi_+ N(\mu_+, \sigma^2)$; Three non-null Gaussian mixture distribution $\pi_- N(\mu_-, \sigma_1^2) + \pi_0 N(0,1) + \pi_+ N(\mu_+, \sigma_2^2) + \pi_{++} N(\mu_{++}, \sigma_3^2)$, where $\pi_- = 2\pi_+ = 2\pi_{++}$; Two non-null $t$-distributed heavy-tailed components $\pi_-[t(1) + \mu_-] + \pi_0 N(0,1) + \pi_+[t(1) + \mu_+]$, where $t(1)$ denotes a $t$-distribution with 1 degree of freedom, and $\pi_- = \pi_+$. $\mu_-$ was randomly drawn from $-|N(1,1)|$, $\mu_+$ and $\mu_{++}$ were drawn from $|N(1,1)|$, and the $\sigma^2$'s were randomly come from $\chi^2(1)$. Five different true values of $\pi_0$, 0.1, 0.3, 0.5, 0.7, and 0.9 were examined. In each scenario, the estimates of all model parameters including $\Theta$ were estimated and compared to the true values. The simulation was repeated for 999 times.

For each true model, the true value of $\Theta$ was determined by knowing the generated Z-scores from their pre-assigned components in the mixture distribution. Denoting a set of independent Z-scores as $\{z_{-1}, \cdots, z_{-k_-}, z_{01}, \cdots, z_{0k_0}, z_{+1}, \cdots, z_{+k_+}\}$, where $k_0 = \pi_0 k$, $k_+ = \pi_+ k$, $k_- = $

$\pi_- k$, and the $+$ and $-$ subscripts indicate that the Z-scores were generated from positive and negative components of the mixture, respectively. The true value of $\Theta$ was calculated as $|\frac{1}{k_-} \sum_{i=1}^{k_-} z_{-i}| + |\frac{1}{k_+} \sum_{i=1}^{k_+} z_{+i}|$, where the $z$'s are the elements of $\mathbf{Z}_-$ and $\mathbf{Z}_+$ vectors, respectively. The true values are comparable across different true models. The true $\Theta$ measures the weighted average non-null genetic effect.

**Simulations with correlated Z-scores.** For the three true models above with Gaussian components, the real phenotypic correlation matrix between 1376 UKBB phenotypes were used to generate 1376 correlated Z-scores for a single genetic variant. We adjusted the phenotypic correlations as described above prior to the TGCA analysis. The simulation was repeated for 999 times.

Similar to the case with independent Z-scores, for each true model, denoting a set of generated correlated Z-scores as $\{z_{-1}, \cdots, z_{-k_-}, z_{01}, \cdots, z_{0k_0}, z_{+1}, \cdots, z_{+k_+}\}$, where $k_0 = \pi_0 k$, $k_+ = \pi_+ k$, $k_- = \pi_- k$, and the $+$ and $-$ subscripts indicate that the Z-scores were generated from positive and negative components of the mixture, respectively. The true value of $\Theta$ was calculated as $|\frac{1}{k_-} \sum_{i=1}^{k_-} z_{-i}^*| + |\frac{1}{k_+} \sum_{i=1}^{k_+} z_{+i}^*|$, where $z^*$'s are the elements of $\mathbf{Z}_-^* = \mathbf{Z}_- \mathbf{R}_-^{-1/2}$ and $\mathbf{Z}_+^* = \mathbf{Z}_+ \mathbf{R}_+^{-1/2}$, and $\mathbf{R}_+$ and $\mathbf{R}_-$ are the correlation matrices for $\mathbf{Z}_-$ and $\mathbf{Z}_+$ vectors, respectively.

**TGCA in the UK Biobank.** We applied the mixture model to the above UK Biobank $\mathbf{Z}^*$ matrix across 2,787,891 QCed SNPs. Standard errors of the parameters were obtained through the Hessian matrix of the likelihood function. The SNPs whose model fitting did not converge in 9999 iterations were filtered out. The SNPs that could not obtain standard errors due to numerically ill-conditioned likelihood surface, i.e. the maximum likelihood estimates could not be properly identified near the boundaries, were filtered out. For each SNP, the standard error of $\hat\Theta$ was calculated via the Delta method (Supplementary Information). We eventually obtained the estimation results of 2,224,394 SNPs for medical conditions, 2,166,115 SNPs for physical measures, 2,366,615 SNPs for mental health, 2,253,831 SNPs for diet, and 2,468,978 SNPs for lifestyle phenotypes across the genome.

**Testing the association between tissues and TGCA.** We downloaded the mean gene-expression data summarised from RNA sequencing by the GTEx project. The GTEx v7 data covers gene expressions of 19,791 genes in 48 human tissues. Gene-expression values were normalised to numbers of transcripts per million reads (TPM) so that they are comparable across tissues. In order to measure the expression specificity of each gene in each tissue, the specificity was defined by dividing the expression of each gene in each tissue by the total expression of that gene in all tissues, leading to specificity ranging from 0 to 1 for each gene.

First, we used stratified LD score regression (S-LDSC)[11, 17] to test whether the 10% most specifically expressed genes in each tissue were enriched for $\hat\Theta$ in each of the five traits domains. The Z-scores of $\hat\Theta$ for each domain of traits were harmonised by the `munge_sumstats.py` procedure of the `ldsc` software. LD scores of HapMap3 SNPs (MHC region excluded) for the annotations of specifically expressed genes in each tissue were computed using a 1-cM window (default). The association between each tissue and $\hat\Theta$ was evaluated by an enrichment score of the proportion of $\hat\Theta$ variability divided by the proportion of annotated SNPs.

Second, similar to the strategy used by the HOPS method[4], we used a SNP-based regression with LD-corrected $\hat\Theta$ to estimate the effect of tissue-specifically expressed genes on $\hat\Theta$, for each tissue and each phenotype domain:

$$\hat\Theta_j = \alpha + \delta \ell_j + \gamma A_j + \epsilon_j \qquad (2)$$

where $\ell_j$ is the LD score of the $j$-th SNP, pre-calculated by the `ldsc` software; $A_j$ takes a value of zero or one, as an indicator for whether the SNP is annotated to be within a tissue-specifically expressed gene; $\gamma$ is the parameter of interest. The more the tissue-specifically expressed genes can explain the variation in $\hat\Theta$, the more positive $\gamma$ would be. As LD exists across the analysed SNPs, directly applying the regression to all the SNPs would underestimate var($\hat\gamma$). We split the SNPs into 100 subsets, where each subset contained SNPs $j, j+100, j+200, \cdots, j = 1, 2, \cdots, 100$, so that the LD correlations were pruned. These resulted in 100 $\hat\gamma$ estimates, and we report the median of them.

**Enrichment of TGCA signals at *cis*-eQTL of tissue-specific genes.** From the GTEx v7 data resource, we extracted the *cis*-eQTL scan summary statistics of the top 100 tissue-specifically expressed genes in each tissue. We represent each *cis*-eQTL with its lead variant, i.e. the SNP with the smallest $p$-value across the tested *cis*-SNPs by both the eQTL scan and our TGCA analysis. We examined the squared Z-score ($\chi^2$) distribution for $\hat\Theta$ in each phenotype domain, stratified on the *cis*-eQTL for the 100 specifically expressed genes in each tissue, where the Z-score is $\hat\Theta / s.e.(\hat\Theta)$. The distribution was evaluated in a quantile-quantile plot against the $\chi^2(1)$ distribution under the null.

**Reporting summary**. Further information on research design is available in the Nature Research Reporting Summary linked to this article.

## Data availability

The whole-genome summary statistics of TGCA for the five phenotype domains are available at the figshare repository https://doi.org/10.6084/m9.figshare.14216324. UK Biobank GWAS summary statistics are available at http://www.nealelab.is/uk-biobank/ukbround2announcement. Gene expression and eQTL data for 48 tissue types are available at: https://www.gtexportal.org/home/datasets. Source data are provided with this paper.

## Code availability

The TGCA source code is available at https://github.com/xiashen/TGCA[18]. The FUMA tool is available at https://fuma.ctglab.nl. The LDSC source code is available at: https://github.com/bulik/ldsc.

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

## Acknowledgements

X.S. was in receipt of a Swedish Research Council (Vetenskapsrådet) Starting Grant (No. 2017-02543). We thank Benjamin Neale's lab for making their UK Biobank GWAS results publicly available. We thank Edinburgh Compute and Data Facility (ECDF) at the University of Edinburgh for providing high-performance computing resources.

## Author contributions

X.S. initiated and coordinated the study; T.L. performed data analysis; T.L., Z.N., and X.S. developed the method; Z.Y., R.Z., C.Z., W.X., K.Y., Y.W., and Y.C. contributed to the analysis pipeline and discussion; T.L. and X.S. drafted the manuscript; Z.N., Z.Y., and R.Z. contributed to manuscript writing.

## Competing interests

The authors declare no competing interests.
