## [Peer Review File · Nature Communications]

Reviewer #1 (Remarks to the Author):

In this manuscript, the authors introduced a novel statistical model to estimate the contribution of a genetic variants to the human phenome. Evaluating the extent of pleiotropy across phenome is one of the challenging yet the most valuable information to understand genetic architecture of human complex traits. As the authors mentioned, most of the existing methods rely on thresholding the significance of the genetic associations which can be problematic due to lack of statistical power or excess of false positives/negatives, while the approach in this manuscript provide a solution to these problems. Overall the manuscript is structured well, however it lacks interpretation of the results and it is not clear how this model can be useful in the field. Since this topic might be an interest of a wide range of readers, it would be appreciated to provide more insight into how this model can contribute to the field. Below are my specific comments.

As the model uses Z score of the genetic associations to a specific trait, the statistical power of the original GWAS does affect the TGCA estimate. As the sample size varies across UKB and majority of the traits are highly polygenic, meaning you need much larger sample size to detect tiny effects, it is possible that the model is biased by the small number of well powered traits (that have relatively large number of traits and less polygenic so the effect sizes are larger). In addition, the authors did not mention if the performance of the model depends on the number of traits considered. These should be at least discussed in the manuscript.

For the analyses or gene-gene interaction and pathways, the findings are interesting, however correlation coefficient is very small (though it's significant). It might be interesting to see how the number of interacting genes/pathways shift by binning the variants based on the significance of the theta estimates. In addition, the authors should provide their insights about what these findings mean.

Given our previous knowledge of wide spread pleiotropy, the significant signals seems very little. The authors should discuss if this is due to lack or power. Is it expected to increase the signals by increasing sample size of the GWASs or the number of phenotypes? Or does this mean the previously reported pleiotropy is largely false positive?

It was not clear from the manuscript how the model can be useful in the field. In addition, the method itself is novel but findings are not. The authors should try to clarify how the results should be interpreted and how can this model be used to further understand the genetic architecture of complex traits. For example, this model is potentially interesting to study overall contribution of variants to traits in a specific domain, e.g. cardiovascular disease or psychiatric disorders. The authors can also extend their analyses to estimate theta for each trait domains.

Reviewer #2 (Remarks to the Author):

In this manuscript, the authors propose a new statistic Theta to assess the overall contribution of a SNP across many traits. They performed simulations to show that their method to estimate Theta yields reasonable estimates of the parameter. They then applied their method to analyze UK Biobank GWAS summary statistics, and observed that brain genes do not show enrichment in TGCA statistic. Overall, I find that the manuscript lacks sufficient description of their method and analysis, and could be significantly improved by much more rigorous simulations. Additionally, the method relies on the adjustment of Z-scores, which seems to have non-unique solutions. Some of the claims regarding brain genes are also likely biased by the set of traits they analyzed.

Major comments:

Overall, there is very little explanation on what the Theta parameter is quantifying. The authors provide a definition of Theta on line 65, but do not offer any further explanation. What does Theta represent? What is the range of Theta? And how does Theta compare, conceptually, with existing metrics of pleiotropy? Since meaning of Theta is crucial to the understanding of this paper, the authors should provide more detailed explanation on what Theta is.

In the "Adjusting phenotypic correlations" subsection of the "Online Methods" section, the authors propose to multiply the Z-score matrix by $R^{-1/2}$ to obtain an adjusted Z-score matrix, which has the property that dot product of two columns of this matrix is non-zero only if the two columns are the same column. I have 2 concerns about this approach. 1) The Z^* matrix is not unique. The authors propose to use eigenvalue decomposition to obtain $R^{-1/2}$. However, one could also obtain such a matrix as follows. Let $R = LL'$ be the Cholesky decomposition of R . And let $Z^* = (L^{-1})'$. Then one can show that $Z^{*'}Z = L^{-1}LL'(L^{-1})' = L^{-1}LL'(L')^{-1} = I$. What would be the appropriate interpretation of the results from analysis of real traits, if the adjusted Z-score matrix, Z^* , is not unique? 2) It is not exactly clear what each row of the adjusted Z-score matrix, Z^* , represents – after multiplying each of Z by $R^{-1/2}$, each row does not represent the association statistics of a SNP across multiple traits, and does not seem to have meaningful interpretation. The authors should provide some intuition about what each row of Z^* represents.

In the "Estimation of phenotypic test statistics correlations" subsection of the "Online Methods" section, the authors argue that the correlation between $\hat{\beta}_{1j}$ and $\hat{\beta}_{2j}$ should be close to the phenotypic correlation, and propose to use correlation of Z-scores of SNPs with minor allele frequency less than $5e-4$ to estimate this quantity. While this approach seems reasonable, I wonder if the authors have also considered using cross-trait LD score regression to estimate this quantity? The intercept term of cross-trait LD score regression incorporates both phenotypic correlation and sample overlap, and seems to be a more principled approach to estimate phenotypic test statistics correlations.

In the "Statistical modeling of TGCA" subsection of the "Online Methods" section, the authors define the TGCA model, assuming effect size of a SNP across multiple traits follows a mixture of Gaussian distributions: a component with negative effect size, a component with null effect size, and a component with positive effect size. Although this is an interesting formulation, there are likely multiple equivalent parameterizations of the mixture model, i.e. the mixture model is likely not unique. For example, a mixture model with high π_{plus} and low μ_{plus} could be equivalent to a mixture model with low π_{plus} and high μ_{plus} . The authors should discuss the possible issues of this modeling in the methods section.

Also, the authors do not provide sufficient details about how to perform statistical inference of the TGCA Theta statistic. The authors should clarify how the standard errors of Theta are exactly obtained, what the null hypothesis is, and what distribution is assumed to obtain the p-value. In addition, the authors should perform null simulations to assess whether their test statistics are well-calibrated at different thresholds, and alternative simulations to assess the power of their test statistics.

In the "Simulations" section, the authors simulated 200 independent phenotypes, and directly fit the TGCA model to obtain the Theta statistics. Here, the authors skipped the step of adjusting the Z-score matrix, which is an important step of their method. Here, I suggest the authors to also perform simulations with correlated phenotypes, to assess whether the Z-score adjustment step produces robust Theta estimates.

It is not exactly clear what the authors did with S-LDSC and TGCA summary statistics. From the description in the subsection "Gene-expression-mediated genetic effects concentrate on non-brain tissues", the authors seem to have run S-LDSC and TGCA summary statistics. This analysis is likely not justified, as the TGCA summary statistics likely violates the assumptions made by the S-LDSC model. In the supplementary information document, the authors seem to have also created an annotation based on the TGCA p-values, and then run S-LDSC on summary statistics of each trait. This procedure uses the summary statistics twice, once for creating the annotation and once for performing S-LDSC analysis, which will likely result in bias. In a more rigorous analysis, the annotation should be created using data independent from the data used to perform S-LDSC analysis.

In the subsection "Gene-expression-mediated genetic effects concentrate on non-brain tissues", the authors claim that their "data preparation process was not subjectively biased towards particular types of traits". This is likely not necessarily the case, as the UK Biobank is overall ill

powered for studying case/control phenotypes, including psychiatric disorders such as schizophrenia and autism. In other words, there are likely very few brain-related traits in the analysis of real traits. This could contribute to the observed lack of enrichment in brain genes. Additionally, the authors should clarify how they obtained p-value assessing the significance of the difference between brain and non-brain enrichment. One concern I have here is that there is strong correlation across different tissues, leading to smaller standard errors.

Minor comments:

In figure 4a, I suggest the authors to show the proportion of SNPs having the annotation right after the annotation name, as it is difficult to see the labels of the bars to the left of the plot.

The contour plot in Figure 2b is difficult to understand. I suggest the authors to consider a more intuitive plot to show the relationship between Theta and P_m/P_n .

Reviewer #3 (Remarks to the Author):

This paper presents a new method for evaluating the pleiotropic contribution of an individual variant to the overall human phenome. While the method is interesting and innovative, and contributes to an important ongoing discussion on pervasive pleiotropy, the score and the model behind it are not explained or justified sufficiently, and as a result I am not entirely sure how to interpret the results.

According to my understanding, the model used in the paper models GWAS associations for multiple traits at a single locus as a mixture of three Gaussians: a null distribution, with mean 0 and variance 1, and two non-null distributions, with means μ_{-} and μ_{+} and variances σ_{-}^2 and σ_{+}^2 . These means and variances are free parameters, which are inferred independently for each SNP, along with the mixture proportions π_{-} , π_0 , and π_{+} . The TGCA statistic Θ , defined as $|\pi_{-}\mu_{-}| + |\pi_{+}\mu_{+}|$, measures the expected absolute value of effect from the combination of the two non-null distributions. Given this understanding, I have the following questions and concerns about the method:

1. Why should we expect effect sizes to be described by a mixture of three and only three Gaussians? It is not at all obvious to me that this is the correct model. Is this a previously established model, or the authors' own invention? In either case it needs to be justified better.
2. What happens to the model and the score if this model is incorrect -- for example, if there is only one non-null component, or if there are three non-null components, or if the non-null components are not Gaussian? The simulations do not address this question, since they use only the three-Gaussian model, and I suspect that the score may not perform well in these cases.
3. What is the null hypothesis tested by the p-values reported in the manuscript? It is not explicitly stated, but my best understanding is that the p-value tests whether Θ is significantly different from zero. In the case where the model is correct, this measures what the authors intend it to, but if the model is incorrect, it may instead measure goodness of fit to the three-Gaussian model, which does not have a straightforward relationship with biological effect or pleiotropy.
4. It is asserted in multiple places in the manuscript that Theta measures both pleiotropy and effect size. However, based on the definition of Θ , it appears to measure only the mean non-null effect size. According to my understanding of the model, the level of pleiotropy (i.e. the number of traits affected) would correspond most closely to the non-null mixture parameters π_{+} and π_{-} . Theta does not measure this quantity, instead reporting the mean effect size across all traits. Is there some reason I am missing why Θ should be interpreted as measuring the level of pleiotropy?

I also have some major comments unrelated to the conceptual properties of the model and the score:

5. The procedure for producing normalized and decorrelated summary statistics -- first excluding extremely highly correlated pairs of traits, then transforming the matrix of Z-scores by the inverse

of its correlation matrix -- is nearly identical to the procedure used by the HOPS method. (For the sake of full disclosure, I am the first author of this paper.) While our method is cited later as a point of comparison to the TGCA method, I was dismayed to see this procedure presented as a novel invention of the authors without citing previous work. It also makes me wonder what other sources and inspirations are uncited in the development of the mixture model and the Θ score.

6. The simulations presented are highly limited in scope, and do not sufficiently establish the ability of the inference method to capture the Theta statistic in a range of circumstances. In particular, the values of σ_{12} , σ_{22} , and Θ itself are held constant through all simulations, and μ_{+} and μ_{-} are guaranteed to be symmetric around 0. I would like to see more variability in all of these parameters, but especially in the value of Θ -- it seems a little strange to claim that estimation of Θ is robust when only one value of Θ was tested.

Despite these shortcomings, I do think this method is extremely interesting and clever and I look forward to seeing a more fully developed description of it.

Daniel M. Jordan

Point-to-point responses to reviewers

for “Total genetic contribution assessment across the human genome”

December 12, 2020

Contents

1	To the reviewers	2
2	Responses to Reviewer #1	2
2.1	Unequal sample sizes across traits	2
2.2	Gene-gene interaction and pathways	3
2.3	Power for detecting pleiotropy	3
2.4	Utility of the model	4
3	Responses to Reviewer #2	5
3.1	The meaning of Θ	5
3.2	On the Z^* matrix	6
3.3	Estimation of phenotypic correlations	6
3.4	Parameterisation of the model	7
3.5	Statistical inference of Θ	7
3.6	Simulation setup	8
3.7	S-LDSC analysis	9
3.8	Tissue enrichment	9
4	Responses to Reviewer #3	10
4.1	Model interpretation	10
4.2	Citations	13
4.3	Additional simulations	14

1 To the reviewers

We would like to thank the three reviewers, for their detailed, constructive, and insightful feedbacks on our work. We have tried our best to address all these comments. Particularly, we have made major revisions to the main figures, in order to fit 1) the new series of simulations, 2) the new sets of results for different trait domains, and 3) the new subsequent analysis. We tend to cover the most interesting points in the main text and put the rest in the supplements.

We believe the paper has been substantially improved, on both technical and practical perspectives. Point-to-point responses to each reviewer are given in the following sections.

2 Responses to Reviewer #1

In this manuscript, the authors introduced a novel statistical model to estimate the contribution of a genetic variants to the human phenome. Evaluating the extent of pleiotropy across phenome is one of the challenging yet the most valuable information to understand genetic architecture of human complex traits. As the authors mentioned, most of the existing methods rely on thresholding the significance of the genetic associations which can be problematic due to lack of statistical power or excess of false positives/negatives, while the approach in this manuscript provide a solution to these problems. Overall the manuscript is structured well, however it lacks interpretation of the results and it is not clear how this model can be useful in the field. Since this topic might be an interest of a wide range of readers, it would be appreciated to provide more insight into how this model can contribute to the field. Below are my specific comments.

2.1 Unequal sample sizes across traits

Q: As the model uses Z score of the genetic associations to a specific trait, the statistical power of the original GWAS does affect the TGCA estimate. As the sample size varies across UKB and majority of the traits are highly polygenic, meaning you need much larger sample size to detect tiny effects, it is possible that the model is biased by the small number of well powered traits (that have relatively large number of traits and less polygenic so the effect sizes are larger). In addition, the authors did not mention if the performance of the model depends on the number of traits considered. These should be at least discussed in the manuscript.

A: The aim of the total genetic contribution estimation is to quantify the overall magnitude of genetic effects across a group of traits. Therefore, for each single SNP, unequal genetic effects across traits are modelled by our mixture model, but indeed, we agree with the referee that unequal sample sizes could generate bias since Z scores are used as data. In order to adjust for the power difference across traits, in the revision, we take into account the sample sizes of the included phenotypes. We re-weight the Z scores per SNP by the square root of the sample sizes, so that the sample

size factor in the standard errors of the genetic effects are removed. After this adjustment, we did see an improvement of genome-wide statistical inference of the estimated total genetic contribution parameter Θ (Fig. 2, Extended Data Fig. 11-12). We found that the number of traits considered in the model does affect the power (Extended Data Fig. 15), nevertheless, for the same true Θ value, the μ parameter seems to affect the power more (as the locations of the mixture components are more apart and identifiable) (Extended Data Fig. 3). We also considered five different types of traits as differently sized phenotype groups. One to two hundreds of related complex traits appear to be effective. In general, a larger number of traits would increase power of the mixture model to identify the parameters. Less polygenic traits, e.g., omics phenotypes, can have major loci with relatively larger effects, which may require less number of phenotypes to identify the mixture components. We added these discussions to the revised manuscript (lines 224-229).

2.2 Gene-gene interaction and pathways

Q: For the analyses of gene-gene interaction and pathways, the findings are interesting, however correlation coefficient is very small (though it's significant). It might be interesting to see how the number of interacting genes/pathways shift by binning the variants based on the significance of the theta estimates. In addition, the authors should provide their insights about what these findings mean.

A: We are glad that the reviewer finds such analysis interesting. We tried the binning strategy but the results did not reveal stronger correlation. In the revised manuscript, we focus on the estimated Θ quantity in different trait domains (following Reviewer 1's suggestion), and we found that such small correlation appeared to be the most significant for mental health traits (Supplementary Fig. 2). We believe the poor correlation could be due to several reasons, including uncertainty in Θ estimates as well as noise (incompleteness) of the interaction and pathway databases. As we used gene-based p-values, it also introduced more uncertainty, and for now unfortunately it is difficult to obtain a gene-based Θ . In the Supplementary Results, we briefly discuss these results. We do not want to emphasise such results too much but rather provide them as potentially insightful points for future investigations.

2.3 Power for detecting pleiotropy

Q: Given our previous knowledge of wide spread pleiotropy, the significant signals seems very little. The authors should discuss if this is due to lack of power. Is it expected to increase the signals by increasing sample size of the GWASs or the number of phenotypes? Or does this mean the previously reported pleiotropy is largely false positive?

A: If we define "pleiotropic loci" as the loci having genome-wide significant effects on more than one phenotype, we do believe that the previously reported pleiotropy is mostly true positive. So the power for detecting reported pleiotropic loci seems to be low if we only look at the p-values of the total genetic contribution parameter Θ . First, as the referee

points out, the power for the inference of Θ and any other parameter in the mixture model increase with both the sample sizes of the GWASs (Extended Data Fig. 16) and the number of phenotypes modelled (Extended Data Fig. 15). Second, Θ is a quantity with its own meaning, and the hypothesis testing against $\Theta = 0$ is not very informative. In the revision, we try to emphasize the value of Θ itself rather than how significant it is compared to zero, as the relative magnitudes of Θ 's across different SNPs are more interesting. On the other hand, *threshold-free* quantification of total genetic effects is a main point of our model. In order to achieve this, it appears although we seem to lose visible power with the mixture modeling, the strategy does highlight some key known loci that have not only widespread pleiotropic but also relatively larger effects on many traits.

For complex traits, it is hard to believe there is any locus that is not pleiotropic. Even with sufficient power, the true Θ 's for most SNPs would be non-zero but equally small, while some loci could be exceptional for particular types of traits – this is indeed what we see in the revision. We believe the threshold-free parameter, regardless of power, provides a fair and unified evaluation across the genome. Similar to GWAS summary statistics of a phenotype, although only a small proportion of loci can be discovered in the GWAS, the whole-genome profile has its own value. We added these discussions to the revised manuscript (lines 230-240).

2.4 Utility of the model

Q: It was not clear from the manuscript how the model can be useful in the field. In addition, the method itself is novel but findings are not. The authors should try to clarify how the results should be interpreted and how can this model be used to further understand the genetic architecture of complex traits. For example, this model is potentially interesting to study overall contribution of variants to traits in a specific domain, e.g. cardiovascular disease or psychiatric disorders. The authors can also extend their analyses to estimate theta for each trait domains.

A: We thank the referee for this valuable feedback, and we agree that applying the model in different trait domains would be more interesting and useful to show the value in applications. In the revision, we considered five different domains of traits as phenotype groups, including physical measures, medical conditions, mental health, lifestyle, and diet. The results highlight two advantages of studying phenotypes within the same domain: 1) As the phenotypes in the same domain share similar genetic architecture, the noise introduced by phenotypes outside the domain was reduced. This results in increased power to identify key loci with high total genetic contribution specifically for the particular domain (Fig. 2, Extended Data Fig. 11-12). 2) Splitting traits into different domains helps interpret the estimated Θ parameter, as well as the π and μ parameters, i.e., how much genetic effect in total a SNP contributes to the traits of the domain, on what proportion of traits in the domain the SNP affects, and what the average effect the SNP has on each trait in the domain. In the Discussion (lines 224-229), we also comment on the future studies using the model, especially on omics phenotypes, where the number of traits is usually large and the key loci with large Θ

may highlight important regulatory hubs in the genome.

3 Responses to Reviewer #2

In this manuscript, the authors propose a new statistic Theta to assess the overall contribution of a SNP across many traits. They performed simulations to show that their method to estimate Theta yields reasonable estimates of the parameter. They then applied their method to analyze UK Biobank GWAS summary statistics, and observed that brain genes do not show enrichment in TGCA statistic. Overall, I find that the manuscript lacks sufficient description of their method and analysis, and could be significantly improved by much more rigorous simulations. Additionally, the method relies on the adjustment of Z-scores, which seems to have non-unique solutions. Some of the claims regarding brain genes are also likely biased by the set of traits they analyzed.

Major comments:

3.1 The meaning of Θ

Q: Overall, there is very little explanation on what the Theta parameter is quantifying. The authors provide a definition of Theta on line 65, but do not offer any further explanation. What does Theta represent? What is the range of Theta? And how does Theta compare, conceptually, with existing metrics of pleiotropy? Since meaning of Theta is crucial to the understanding of this paper, the authors should provide more detailed explanation on what Theta is.

A: We apologise for unclear interpretation of Θ in the original manuscript, especially when it comes to the definition of “pleiotropy” in general. Different from an ordinary pleiotropy concept, which one might simply define as a SNP affecting more than one trait, the Θ parameter aims to quantify the total genetic contribution of each SNP on a group of phenotypes. Its main advantage is being *threshold-free*, which is needed in the field of quantitative genetics. Θ is formed with π and μ parameters, together helping us interpret how much genetic effect in total a SNP contributes to the traits of the domain, on what proportion of traits in the domain the SNP affects, and what the average effect the SNP has on each trait in the domain. In the revision, we clarified these points when defining the Θ parameter (lines 68-77).

Also, to assist interpretation of Θ , in the revision, taking a suggestion from Reviewer 1, we considered five different domains of traits as phenotype groups, including physical measures, medical conditions, mental health, lifestyle, and diet. This results in not only increased power to identify key loci with high total genetic contribution specifically for the particular domain (Fig. 2), but also helps interpret the estimated Θ parameter, as well as the π and μ parameters in the context of each trait domain.

3.2 On the Z^* matrix

Q: In the “Adjusting phenotypic correlations” subsection of the “Online Methods” section, the authors propose to multiply the Z-score matrix by $R^{-1/2}$ to obtain an adjusted Z-score matrix, which has the property that dot product of two columns of this matrix is non-zero only if the two columns are the same column. I have 2 concerns about this approach. 1) The Z^* matrix is not unique. The authors propose to use eigenvalue decomposition to obtain $R^{-1/2}$. However, one could also obtain such a matrix as follows. Let $R = LL'$ be the Cholesky decomposition of R. And let $Z^* = (L^{-1})'$. Then one can show that $Z^{*'}Z = L^{-1}LL'(L^{-1})' = L^{-1}LL'(L')^{-1} = I$. What would be the appropriate interpretation of the results from analysis of real traits, if the adjusted Z-score matrix, Z^* , is not unique? 2) It is not exactly clear what each row of the adjusted Z-score matrix, Z^* , represents – after multiplying each of Z by $R^{-1/2}$, each row does not represent the association statistics of a SNP across multiple traits, and does not seem to have meaningful interpretation. The authors should provide some intuition about what each row of Z^* represents.

A: We thank the reviewer for bringing up the non-unique solutions of Z^* . First of all, we need to apologise for not citing the HOPS paper when it comes to creating the Z^* matrix – The procedure was also used in the HOPS pipeline to decorrelate the Z-scores across traits, and we adopted the same strategy. The reasoning behind such a decorrelation step is to create uncorrelated Z-scores per SNP, so that the inference of mixture model becomes straightforward. Although there are non-unique solutions of Z^* , as long as it’s a linear transformation of Z , the procedure simply redistributes most of the estimated genetic effects onto the leading eigenvectors. Therefore, as the reviewer points out, each single value in the Z^* matrix does not represent an association statistic on an original phenotype anymore, but rather an “association statistic” for a linear combination of many original phenotypes. This is also why, in the context and aim of this work, trying to identify a total genetic contribution parameter Θ would be more meaningful (and identifiable), rather than identifying which phenotypes the SNP affects. We think these can be useful discussion for some readers, so we include in the revised Discussion (lines 241-248).

3.3 Estimation of phenotypic correlations

Q: In the “Estimation of phenotypic test statistics correlations” subsection of the “Online Methods” section, the authors argue that the correlation between $\hat{\beta}_{1j}$ and $\hat{\beta}_{2j}$ should be close to the phenotypic correlation, and propose to use correlation of Z-scores of SNPs with minor allele frequency less than $5e-4$ to estimate this quantity. While this approach seems reasonable, I wonder if the authors have also considered using cross-trait LD score regression to estimate this quantity? The intercept term of cross-trait LD score regression incorporates both phenotypic correlation and sample overlap, and seems to be a more principled approach to estimate phenotypic test statistics correlations.

A: We wanted to explain our logic on the estimation of phenotypic correlations in this paper, as we thought this is an established procedure. Later on, during our further investigation, we realised that it is indeed worthwhile to explain the theory behind estimating such correlations via correlating $\hat{\beta}$'s or Z 's between phenotypes. In the end, we wrote another manuscript on our phenotypic estimation procedure, which we posted on a preprint server for both the reviewers and peers interested in this topic (<https://www.biorxiv.org/content/10.1101/2020.12.10.419325v1>). To summarise, our “low-MAF” method is not only simple but also more effective than the state-of-the-art strategies including LD score regression intercept, as the intercept actually carries biases due to population substructure etc. We cited this preprint in the revised manuscript.

3.4 Parameterisation of the model

Q: In the “Statistical modeling of TGCA” subsection of the “Online Methods” section, the authors define the TGCA model, assuming effect size of a SNP across multiple traits follows a mixture of Gaussian distributions: a component with negative effect size, a component with null effect size, and a component with positive effect size. Although this is an interesting formulation, there are likely multiple equivalent parameterizations of the mixture model, i.e. the mixture model is likely not unique. For example, a mixture model with high π_+ and low μ_+ could be equivalent to a mixture model with low π_+ and high μ_+ . The authors should discuss the possible issues of this modeling in the methods section.

A: We thank the reviewer for this technical comment. Obviously, a SNP with high π_+ and low μ_+ could have the same Θ as a SNP with low π_+ and high μ_+ . This can apply to both the estimated Θ and the true Θ , so indeed the model-estimated Θ itself cannot distinguish the two types of SNPs. The estimated π and μ parameters can help distinguish these two different scenarios. Nevertheless, for the same true value of Θ , the power to identify Θ varies across different π - μ combinations – the power is higher for low π_+ and high μ_+ as the locations of the mixture components are more apart and identifiable. We added these points to the revised Discussion (lines 208-214).

3.5 Statistical inference of Θ

Q: Also, the authors do not provide sufficient details about how to perform statistical inference of the TGCA Theta statistic. The authors should clarify how the standard errors of Theta are exactly obtained, what the null hypothesis is, and what distribution is assumed to obtain the p-value. In addition, the authors should perform null simulations to assess whether their test statistics are well-calibrated at different thresholds, and alternative simulations to assess the power of their test statistics.

A: We apologise for not providing clear description regarding the inference of Θ and the related p-values. To address this, first, we would like to emphasise that pleiotropy is very widespread across the genome (as Reviewer 1 also points out). It is hard to believe that any locus in the genome has a completely zero effect on any given phenotype. Even

with great statistical power, the true Θ 's for most SNPs would be non-zero but equally small, while some loci could be exceptional for particular types of traits – which is indeed what we see in the revision. Therefore we found it not particularly interesting to test against the null hypothesis of $\Theta = 0$, which is what our p-values were for in the original manuscript. In the revision, we try to emphasise the value of Θ itself rather than how significant it is compared to zero, as the relative magnitudes of Θ 's across different SNPs are more interesting, and quantification of the total genetic contribution (as a score) is the aim of the parameter, rather than whether such a contribution is non-zero. The uncertainty of the estimated genetic effects in GWAS due to allele frequency and sample size is already considered in the Z-scores, so that across SNPs, the contrast in Θ is already useful. Nevertheless, the standard error can be useful when the readers do need a p-value, so as the reviewer suggested, we did a series of simulations to assess the validity of our p-values and report the operating characteristics curves (Extended Data Fig. 3, 15, 16). As it is a Wald test, the standard error for each parameter was obtained based on a quadratic approximation to the likelihood function. Such an approximation usually works well when the likelihood function is or close to a regular shape around its maximum. We found that under the null, the p-value distribution is close to uniform though slightly deviates at a cutoff around .05 to .1. Such a null distribution is acceptable especially for the SNPs in our real data analysis that have relatively large Θ estimates. In general, the power grows when the number of traits increases, and for the same true value of Θ , the power is higher for e.g., low π_+ and high μ_+ as the locations of the mixture components are more apart and identifiable. We added these points to the revised Discussion (lines 208-214).

3.6 Simulation setup

Q: In the “Simulations” section, the authors simulated 200 independent phenotypes, and directly fit the TGCA model to obtain the Theta statistics. Here, the authors skipped the step of adjusting the Z-score matrix, which is an important step of their method. Here, I suggest the authors to also perform simulations with correlated phenotypes, to assess whether the Z-score adjustment step produces robust Theta estimates.

A: We thank the reviewer for this constructive suggestion. In addition to independent setups, in the revised manuscript, for the main simulations and applications, we included scenarios with correlated phenotypes, where the correlation structure comes from the real observed correlation structure of the UK Biobank phenotypes. Top eigenvectors explaining 90% information of the correlation matrix were passed onto subsequent simulation and computational steps. Also, following a suggestion by Reviewer 3, we develop the simulation to be more flexible when it comes to model specification and parameter values. We made the true values of the μ 's, σ^2 's, and Θ all drawn from random distributions, so that a wide range of parameter combinations are considered. In this way, the simulation results cannot be displayed as the original boxplots, instead, we visualise the distance to true Θ values for estimation consistency and efficiency (Fig. 1, Extended Data Fig. 1-2).

3.7 S-LDSC analysis

Q: It is not exactly clear what the authors did with S-LDSC and TGCA summary statistics. From the description in the subsection “Gene-expression-mediated genetic effects concentrate on non-brain tissues”, the authors seem to have run S-LDSC and TGCA summary statistics. This analysis is likely not justified, as the TGCA summary statistics likely violates the assumptions made by the S-LDSC model. In the supplementary information document, the authors seem to have also created an annotation based on the TGCA p-values, and then run S-LDSC on summary statistics of each trait. This procedure uses the summary statistics twice, once for creating the annotation and once for performing S-LDSC analysis, which will likely result in bias. In a more rigorous analysis, the annotation should be created using data independent from the data used to perform S-LDSC analysis.

A: We thank the reviewer for pointing out this concern, which was something we also wondered ourselves when running the S-LDSC pipeline. For the first part of the question, indeed that the Z-scores of $\hat{\Theta}$ do not have the same distribution as the Z’s from GWAS summary statistics. Nevertheless, the correlation structure serves as the sufficient statistic for estimating an underlying “heritability” and subsequent enrichment (see the full likelihood of GWAS Z’s in our recent report: <https://www.nature.com/articles/s41588-020-0653-y>). Namely, in order to justify the usage of S-LDSC for enrichment analysis, we basically need the correlation structure of the Z-scores of $\hat{\Theta}$ to be similar to that of the GWAS Z’s, which is the LD correlation structure. Thus, in the revision, we examined the relationship between the correlation of the Z-scores of $\hat{\Theta}$ and the LD correlation between two SNPs (Extended Data Fig. 13). The correlation values do not follow identity but do follow a monotonic trend, so we expect S-LDSC can be used to test enrichment of $\hat{\Theta}$ though the heritability values won’t have clear interpretation. We also tried to avoid S-LDSC when it comes to examining enrichment of tissue specific genes and genome annotations (Fig. 3, Supplementary Fig. 3). In the revised manuscript, we adopted the LD-correction strategy used by the HOPS paper, to test the enrichment of $\hat{\Theta}$ on each particular annotation (set of SNPs). LD scores were used as covariate to remove variation in $\hat{\Theta}$ due to LD difference across SNPs. We also applied some simple LD pruning to try to remove potential bias in enrichment assessment due to LD. For the main results (Fig. 3), we compared the scores from the two strategies, so that results with agreement between the methods would be more trustworthy.

3.8 Tissue enrichment

Q: In the subsection “Gene-expression-mediated genetic effects concentrate on non-brain tissues”, the authors claim that their “data preparation process was not subjectively biased towards particular types of traits”. This is likely not necessarily the case, as the UK Biobank is overall ill powered for studying case/control phenotypes, including psychiatric disorders such as schizophrenia and autism. In other words, there are likely very few brain-related traits in the analysis of real traits. This could contribute to the observed lack of enrichment in

brain genes. Additionally, the authors should clarify how they obtained p-value assessing the significance of the difference between brain and non-brain enrichment. One concern I have here is that there is strong correlation across different tissues, leading to smaller standard errors.

A: We agree with the reviewer regarding these raised points. First, following a suggestion by Reviewer 1, in the revision, we considered five different domains of traits as phenotype groups, including physical measures, medical conditions, mental health, lifestyle, and diet. As the phenotypes in the same domain share similar genetic architecture, the noise introduced by phenotypes outside the domain was reduced. This results in increased power to identify key loci with high total genetic contribution specifically for the particular domain (Fig. 2). With these, we observed different patterns of tissue enrichment for different trait domains, e.g., the brain tissues does appear to be more connected to mental health traits (Fig. 3). We skipped the meta-analysis of brain tissues v.s. non-brain tissues, due to the concern in correlations between tissues.

Q: Minor comments:

In figure 4a, I suggest the authors to show the proportion of SNPs having the annotation right after the annotation name, as it is difficult to see the labels of the bars to the left of the plot.

A: In the revision, we moved the display of annotation enrichment to Supplementary Figure 3. The format changed but we still follow the reviewer’s suggestion to label the proportions of SNPs together with the annotation names.

Q: The contour plot in Figure 2b is difficult to understand. I suggest the authors to consider a more intuitive plot to show the relationship between Theta and P_m/P_n .

A: In the revision, we provide different ways to visualise the difference across the different statistics (Supplementary Fig. 5-9), including a direct comparison of the values between $\hat{\Theta}$ and the other statistics.

4 Responses to Reviewer #3

This paper presents a new method for evaluating the pleiotropic contribution of an individual variant to the overall human phenome. While the method is interesting and innovative, and contributes to an important ongoing discussion on pervasive pleiotropy, the score and the model behind it are not explained or justified sufficiently, and as a result I am not entirely sure how to interpret the results.

4.1 Model interpretation

Q: According to my understanding, the model used in the paper models GWAS associations for multiple traits at a single locus as a mixture of three Gaussians: a null distribution, with mean 0 and variance 1, and two non-null distributions, with means μ_- and μ_+ and variances σ_1^2 and σ_2^2 . These means and variances are free parameters,

which are inferred independently for each SNP, along with the mixture proportions π_- , π_0 , and π_+ . The TGCA statistic Θ , defined as $|\pi_- \mu_-| + |\pi_+ \mu_+|$, measures the expected absolute value of effect from the combination of the two non-null distributions. Given this understanding, I have the following questions and concerns about the method: 1. Why should we expect effect sizes to be described by a mixture of three and only three Gaussians? It is not at all obvious to me that this is the correct model. Is this a previously established model, or the authors' own invention? In either case it needs to be justified better.

A: We would like to apologise that we missed a citation to our own work last year in *Frontiers in Genetics* (<https://www.frontiersin.org/articles/10.3389/fgene.2019.00669/full>). It was cited in an earlier version of the paper, but later we moved the reference to Discussion but somehow wasn't typeset. In that short commentary, although we did not formally justify the properties of such a mixture model, we did "invent" it to estimate the proportion of traits that a variant potentially have effects on. We don't want to emphasise that this is an invention as mixture modeling is flexible, and others may come up with different mixture setup which could answer similar scientific questions. We revised the relevant discussions and included the missing reference (lines 203-207).

To answer the reviewer's first question, we definitely do not necessarily expect that the effects follow such a model in practice. In fact, we do not know the true model. However, there are two advantages of the three-Gaussian setup: 1) The parameters are straightforward to interpret, such as on what proportion of the traits the SNP affects (the π 's), and what the average effect the SNP has on each trait (the μ 's), and how much genetic effect in total a SNP contributes to the traits (Θ); 2) The Gaussian mixture is easier to handle computationally in an EM algorithm, numerically easier to identify the parameters. We included these points in the revised Discussion. Here, we would like to emphasise again that *threshold-free* quantification of total genetic effects is a main point of our model. Based on our work, we believe there can be further development in the modeling that fit the data better or with different interpretable parameters. While we think the current model we propose is sufficient for this paper to illustrate the idea and has its own value and impact. We added these discussions to the revised manuscript (lines 215-223).

Q: 2. What happens to the model and the score if this model is incorrect – for example, if there is only one non-null component, or if there are three non-null components, or if the non-null components are not Gaussian? The simulations do not address this question, since they use only the three-Gaussian model, and I suspect that the score may not perform well in these cases.

A: We thank the reviewer for bringing up this constructive concern, and in the revision, we conducted a series of simulations, exactly following the alternative true models proposed by the reviewer – We simulated four scenarios of true models: two non-null components, one non-null component, three non-null components, and two non-null non-Gaussian components (heavy-tailed t-distributions). Similar to the reviewer, we also suspected that the performance would drop when the model is mis-specified, while the simulation results seem to be interesting and more informative than our expectation: 1) In terms of the estimation of Θ , mis-specification of the model does not affect the consistency

in the estimation, however, for three non-null Gaussian and two non-null fat-tailed t cases, the estimation efficiency drops (Fig. 1, Extended Data Fig. 1-2). Note here that the fat-tailed t-distribution with 1 degree of freedom is a very extreme case. 2) In terms of the inference of Θ , with the same true value, for one and three non-null Gaussian cases, the mis-specification does not affect the identification power for Θ , or sometimes the power can be even higher (Extended Data Fig. 3). This is because for the same true value of Θ , when the non-null proportions are low, the absolute μ 's are high, so that the locations of the mixture components are more apart and identifiable. For the two non-null 1-df t-distribution case, Θ becomes very hard to identify when the null proportion is low. Along with the revised results, we added descriptions about the new simulations in the revised Methods and Discussion.

Q: 3. What is the null hypothesis tested by the p-values reported in the manuscript? It is not explicitly stated, but my best understanding is that the p-value tests whether Θ is significantly different from zero. In the case where the model is correct, this measures what the authors intend it to, but if the model is incorrect, it may instead measure goodness of fit to the three-Gaussian model, which does not have a straightforward relationship with biological effect or pleiotropy.

A: First, we would like to emphasise that pleiotropy is very widespread across the genome (as Reviewer 1 also points out). It is hard to believe that any locus in the genome has a completely zero effect on any given phenotype. So even with great statistical power, the true Θ 's for many SNPs would be non-zero but equally small, while some loci could be exceptional for particular types of traits – which is indeed what we see in the revision. Therefore we found it not particularly interesting to test against the null hypothesis of $\Theta = 0$, which is what our p-values were for in the original manuscript. In the revision, we try to emphasise the value of Θ itself rather than how significant it is compared to zero, as the relative magnitudes of Θ 's across different SNPs are more interesting, and quantification of the total genetic contribution (as a score) is the aim of the parameter, rather than whether such a contribution is non-zero. The uncertainty of the estimated genetic effects in GWAS due to allele frequency and sample size is already considered in the Z-scores, so that across SNPs, the contrast in Θ is already useful. Nevertheless, the standard error can be useful when the readers do need a p-value, and as the reviewer says, it does suggest some sort of goodness of fit. As Reviewer 2 asked for, we did a series of simulations to assess the validity of our p-values and report the operating characteristics curves (Extended Data Fig. 3, 15, 16). As it is a Wald test, the standard error for each parameter was obtained based on a quadratic approximation to the likelihood function. Such an approximation usually works well when the likelihood function is or close to a regular shape around its maximum. We found that under the null, the p-value distribution is close to uniform though slightly deviates at a cutoff around .05 to .1. Such a null distribution is acceptable especially for the SNPs in our real data analysis that have relatively large Θ estimates. We added these points to the revised Discussion (lines 230-240).

Q: 4. It is asserted in multiple places in the manuscript that Theta measures both pleiotropy and effect size. However, based on the definition of Θ , it appears to measure only the mean non-null effect size. According to my understanding of the model, the level of pleiotropy (i.e. the number of traits affected) would correspond most closely to the non-null mixture parameters π_+ and π_- . Theta does not measure this quantity, instead reporting the mean effect size across all traits. Is there some reason I am missing why Θ should be interpreted as measuring the level of pleiotropy?

A: We feel we need to apologise for any misunderstanding our writing generated when it comes to the definition and interpretation of Θ . We tried to clarify these and be more careful with wording in the revision. As the reviewer says, “the level of pleiotropy” can be defined as the number of associated traits for the SNP, or the proportion of traits associated within a certain domain. So we want to emphasise again that quantifying the *total genetic contribution (effect)* is the aim of the paper, the model, and the parameter Θ . It indeed has different properties from the current statistics measuring the level of pleiotropy (Extended Data Fig. 5-9). We tried to avoid saying that Θ is the level of pleiotropy, although it’s a quantity that is certainly affected by the level of pleiotropy.

4.2 Citations

Q: I also have some major comments unrelated to the conceptual properties of the model and the score: 5. The procedure for producing normalized and decorrelated summary statistics – first excluding extremely highly correlated pairs of traits, then transforming the matrix of Z-scores by the inverse of its correlation matrix – is nearly identical to the procedure used by the HOPS method. (For the sake of full disclosure, I am the first author of this paper.) While our method is cited later as a point of comparison to the TGCA method, I was dismayed to see this procedure presented as a novel invention of the authors without citing previous work. It also makes me wonder what other sources and inspirations are uncited in the development of the mixture model and the Θ score.

A: We sincerely apologise for not citing the reviewer’s HOPS work when introducing the decorrelation procedure. We had no intention to claim that the decorrelation procedure was novel in our paper. We simply felt that the main message of the HOPS work is the developed statistics, whereas decorrelation was a technical step in data preparation. In the revision, we specifically cited the HOPS work in the revision when introducing the decorrelation step. Also, we much appreciate the LD-corrected pleiotropy score strategy in HOPS, so when we tried to address the comment on LD score regression enrichment analysis raised by Reviewer 2, we also adopted a similar idea to test for enrichment of Θ with LD correction, where we also cited HOPS. We hope the reviewer won’t be dismayed anymore, and we believe, with the reviewers’ comments and replies published together with the paper, the readers would understand the connection between the development of HOPS and TGCA better.

4.3 Additional simulations

Q: 6. The simulations presented are highly limited in scope, and do not sufficiently establish the ability of the inference method to capture the Theta statistic in a range of circumstances. In particular, the values of σ_1^2 , σ_2^2 , and Θ itself are held constant through all simulations, and μ_+ and μ_- are guaranteed to be symmetric around 0. I would like to see more variability in all of these parameters, but especially in the value of Θ – it seems a little strange to claim that estimation of Θ is robust when only one value of Θ was tested.

A: We agree with the reviewer that the original simulation setup did not cover a wide range of parameter scenarios. In the revision, we develop the simulation to be more flexible when it comes to model specification and parameter values (Fig. 1, Extended Data Fig. 1-2). We made the true values of the μ 's, σ^2 's, and Θ all drawn from random distributions, so that a wide range of parameter combinations are considered, e.g., the μ 's won't be symmetric around zero. In this way, the simulation results cannot be displayed as the original boxplots, instead, we visualise the distance to true Θ values for estimation consistency and efficiency.

Despite these shortcomings, I do think this method is extremely interesting and clever and I look forward to seeing a more fully developed description of it.

Daniel M. Jordan

We strongly acknowledge and appreciate that the reviewer signed this report – an effort that makes the reviewing process more open and helps scientific exchange. We thank the reviewer's HOPS work which provided useful insights in our modeling development.

Dr. Xia Shen, on behalf of the co-authors

Reviewer #1 (Remarks to the Author):

The authors have sufficiently answered my concerns, and I really appreciate nicely organized response letter, it was much easier to follow.

Reviewer #2 (Remarks to the Author):

The authors have put effort into addressing my comments. However, most of my concerns are not addressed adequately enough.

Major comments:

1. It seems that the authors did not highlight their changes in a different color or font. This makes reviewing the revised manuscript quite inconvenient.

2. In the revised manuscript, the authors clarified the meaning of Theta. However, I still have a few concerns about the TGCA method.

2a. What would happen to TGCA if traits have drastically different levels of heritability? The TGCA method does not standardize GWAS Z-scores based on heritability of the traits, although the authors did standardize GWAS Z-scores by the square root of sample sizes.

2b. Why is the variance of the Gaussian distribution of the penalization term, $\pi_0 \cdot N(0, 1)$, set to 1? Setting the variance to 1 is logical if the GWAS Z-scores are *not* standardized by square root of sample sizes, as this is the variance of Z-scores of null SNPs. However, since the authors used Z-scores standardized by square root of sample sizes, shouldn't the variance term of the Gaussian distribution be replaced by $1/(\text{sample size})$?

3. It would be greatly helpful if the authors could theoretically prove and/or empirically demonstrate that TGCA would yield the same results, using either eigenvalue decomposition or Cholesky decomposition. I believe it is the authors' responsibility to show that their method is robust under different transformations of the GWAS summary statistics.

4. I appreciate the authors for dedicating a separate bioRxiv pre-print to investigate best approaches to estimate phenotypic correlation. However, I have to point out that that manuscript has not been peer reviewed. It would be greatly helpful if the authors could demonstrate, within the scope of this current manuscript, that the TGCA method is not affected by different methods of estimating phenotypic correlation.

5a. What would be a threshold for Theta, above which one should be interested in following up on a SNP? I agree with the authors that it is very likely that any random region of the genome affects some complex trait. So, testing if Theta is greater than 0 seems to be a less interesting target. However, testing if Theta is greater than 0 does provide a baseline. The authors mentioned that they would emphasize the magnitude of Theta. However, they did not provide a threshold for Theta.

5b. Also, I am not sure if it is justifiable to say that Z-scores already accounts for allele frequencies and sample sizes. In particular, the authors used Z-scores standardized by sample sizes.

6. I appreciate the authors for perming the additional simulations. My only comment is to add a horizontal line at $\gamma=0$.

7. In the authors' response to my concerns regarding S-LDSC analysis, the authors satisfactorily addressed my first concern regarding the distribution of TGCA Z-scores. However, the author did not seem to address my 2nd concern of using GWAS summary statistics twice, once for creating the annotations, and once for enrichment analysis.

Reviewer #3 (Remarks to the Author):

In general, I am very satisfied with the authors' response to my comments. I particularly want to thank the authors for their positive and friendly responses to my request that my work be cited and to my signing my review. I strongly agree that this kind of exchange is good for the peer review process and for science generally. On a more personal level, in the past I have typically only signed my reviews when my identity is relevant to my comments (as it was here) and it often makes me nervous to do so. Getting a specific positive response to signing my review will make me more willing to sign my reviews in the future, and it is greatly appreciated.

The manuscript is substantially stronger and easier to read now as a result of the responses to mine and the other reviewers' comments. It is much clearer which portions are novel to this work and what the rationale is for the methodological choices the authors have made, and this also makes the sections of the manuscript describing applications of the new method much more compelling. I have the following minor comments as follow-ups to my original comments.

1. I am completely satisfied with the responses to the issues of model choice and robustness. The expanded simulations in particular are extremely compelling, and like the authors I am pleasantly surprised to see that the three-Gaussian model does seem to have general power to fit a large range of alternative models. My only real comment on these simulations is that the text should be clearer about how the "true Θ " is defined in these alternative models, and a brief comment on whether those Θ s are really equivalent across models (though I think it's probably correct that they are). The one thing I would still like to see simulations of is more heterogeneous values of the n parameters. However, the simulations now do cover a very wide range of parameters, and the three-Gaussian model does have some heterogeneity in the n parameters, so I don't think adding additional simulations is necessary for publication.

2. I think the interpretation of Θ is a little clearer, and reading through the revised manuscript I no longer get the impression that it is somehow supposed to measure level of pleiotropy. I still think the idea of a "total genetic contribution" can be explained a little more clearly, to make it clear to readers who can't follow the math that the goal is to get a kind of combined GWAS association score that measures the aggregate effect of a SNP across multiple traits. This probably only requires one additional sentence in the introduction or conclusion repeating this in plain language.

3. I am satisfied with the response to my comment about the definition of p-values (that $\Theta=0$ is an interesting null hypothesis to test but that ultimately the Θ score itself is more important than the p-value) but that response is only reflected in a few of the many places in the revised manuscript where p-values are mentioned. I won't suggest that the p-values be removed from the manuscript entirely, but it might be helpful to change the way they are referenced, so that they are referred to throughout the manuscript as p-values for the null hypothesis of $\Theta=0$, rather than as p-values for the Θ statistic. The most egregious use of these p-values is in Figure 4, where the p-values are used to describe the distribution of Θ scores and the Θ scores themselves are not shown. If possible, that figure and that analysis should be reworked so that the distributions are displayed as Θ values rather than p-values.

Overall, again, I think the manuscript is very much improved and I'm very impressed with the work the authors have put into addressing my comments and the comments of the other reviewers. I look forward to seeing the final version and to seeing future work on this subject.

Daniel M. Jordan

Point-to-point responses to reviewers

for “Total genetic contribution assessment across the human genome” (Revision 2)

January 29, 2021

Contents

1	Response to Reviewer #1	1
2	Responses to Reviewer #2	2
2.1	Highlighting changes	2
2.2	Technical concerns on TGCA modeling and sample size	2
2.3	Decomposition algorithms of the phenotypic correlation matrix	3
2.4	Estimating phenotypic correlations	4
2.5	Threshold for Θ	4
2.6	Visualisation of simulation results	4
2.7	On the S-LDSC analysis	5
3	Responses to Reviewer #3	5
3.1	Defining “true Θ ”	5
3.2	Plain definition of Θ	6
3.3	P-values for the null hypothesis $\Theta = 0$	6
3.4	Final remarks	7

1 Response to Reviewer #1

The authors have sufficiently answered my concerns, and I really appreciate nicely organized response letter, it was much easier to follow.

Thanks for the nice final remark.

2 Responses to Reviewer #2

The authors have put effort into addressing my comments. However, most of my concerns are not addressed adequately enough.

Major comments:

2.1 Highlighting changes

Q1: It seems that the authors did not highlight their changes in a different color or font. This makes reviewing the revised manuscript quite inconvenient.

A1: We apologise that we did not highlight the changes in the first revision, though we did provide in the response letter the line numbers of major changes. In this revision, we provide a text comparison document that highlights the changes we made for the main text.

2.2 Technical concerns on TGCA modeling and sample size

In the revised manuscript, the authors clarified the meaning of Theta. However, I still have a few concerns about the TGCA method.

Q2a: What would happen to TGCA if traits have drastically different levels of heritability? The TGCA method does not standardize GWAS Z-scores based on heritability of the traits, although the authors did standardize GWAS Z-scores by the square root of sample sizes.

Q2b: Why is the variance of the Gaussian distribution of the penalization term, $\pi_0 N(0, 1)$, set to 1? Setting the variance to 1 is logical if the GWAS Z-scores are *not* standardized by square root of sample sizes, as this is the variance of Z-scores of null SNPs. However, since the authors used Z-scores standardized by square root of sample sizes, shouldn't the variance term of the Gaussian distribution be replaced by $1/(\text{sample size})$?

Q5b: Also, I am not sure if it is justifiable to say that Z-scores already accounts for allele frequencies and sample sizes. In particular, the authors used Z-scores standardized by sample sizes.

A2a: We thank the reviewer for raising the question on the levels of heritability, however, we do not think it is a point that should be considered in the current model. There are two main reasons: 1) TGCA models the genetic effects across multiple phenotypes of each single SNP, so only the genetic effects by a single SNP is considered. Thus, regardless of the heritability of each phenotype, whether or not the SNP has an effect on a particular trait depends on the genetic architecture of the trait. For instance, if the trait is highly polygenic such as human height, each SNP may contribute a tiny effect; while if the trait is the expression of a gene, a SNP at the cis-regulatory region may have a much larger genetic effect, though the overall heritability of the gene expression could be much lower than complex

trait such as height. 2) If higher heritability for a trait leads to larger genetic effects (β 's) across the genome, it will make the μ and Θ estimates larger in general. This is what we expect, as heritability affects the true parameter size rather than the estimates. The mixture model already assumes different genetic effects for different traits.

A2b: We understand the referee's concern here. We would like to note that when standardising the sample size, we did not simply divide all the Z-scores for a trait by the square root of its sample size, but rather divide by a weight of $\sqrt{N_j}/(\frac{1}{k} \sum_{i=1}^k \sqrt{N_i})$ for the j -th trait. In this way, the average weight across the traits is 1, so that the average variance of the GWAS Z-scores is 1 under the null. Indeed if the model can consider different sample sizes in the variance of the null component, it would in theory provide a better fit; however, the model would become too complicated to be identified in a mixture model setup. As the expected Z-score variance is 1 under the null, modeling the null component as $N(0, 1)$ is reasonable. We also add this discussion in the revised manuscript (lines 232-237).

A5b: We moved this comment up to this section as its also related to sample size. It looks like there was a misunderstanding. The referee seems to refer to our answer in Section 3.5 of the last response letter, where we stated "The uncertainty of the estimated genetic effects in GWAS due to allele frequency and sample size is already considered in the Z-scores, so that across SNPs, the contrast in Θ is already useful." We apologise if we were not clear enough there. Indeed we standardised the sample sizes to make the Z-scores comparable across the *phenotypes*; however, in this statement, we were trying to discuss whether the Z-scores are comparable across the *SNPs*. Basically, if due to genotyping or imputation failure, the genotypes of the GWASed SNPs have different sample sizes, this is already taken into account in the standard errors of the genetic effect estimates, just as the allele frequencies.

2.3 Decomposition algorithms of the phenotypic correlation matrix

Q3: It would be greatly helpful if the authors could theoretically prove and/or empirically demonstrate that TGCA would yield the same results, using either eigenvalue decomposition or Cholesky decomposition. I believe it is the authors' responsibility to show that their method is robust under different transformations of the GWAS summary statistics.

A: In this revision, we have conducted a set of simulations to demonstrate that the target parameter can still be reasonably consistently estimated if replacing the eigen-decomposition algorithm with Cholesky decomposition (Extended Data Fig. 17). However, we would like to emphasise that for a large number of traits or an ill-conditioned phenotypic correlation matrix (after removing highly correlated phenotypes), one needs to regularise the phenotypic correlation matrix by taking the top eigenvectors (Extended Data Fig. 18) so that the noise gets penalised. Eigen-decomposition enables such regularisation, whereas Cholesky decomposition cannot do so – Cholesky decomposition is normally implemented via a sequential algorithm starting from the first element of the matrix, thus different ordering of the phenotypes in the correlation matrix would result in different decomposed matrix. For a symmetric correlation matrix, singular value decomposition (SVD) can also be used, equivalent to eigen-decomposition as the singular values

and eigenvalues are matched. We added this point in the revised Discussion (lines 263-268).

2.4 Estimating phenotypic correlations

Q4: I appreciate the authors for dedicating a separate bioRxiv pre-print to investigate best approaches to estimate phenotypic correlation. However, I have to point out that that manuscript has not been peer reviewed. It would be greatly helpful if the authors could demonstrate, within the scope of this current manuscript, that the TGCA method is not affected by different methods of estimating phenotypic correlation.

A: In this revision, using the 122 traits in the medical condition trait domain, we demonstrate that the LDSC-intercept-estimated phenotypic correlations are almost the same as those estimated by our simple low-MAF method (Extended Data Fig. 19), so that the post-decorrelation results would not vary. Nevertheless, we also discuss (lines 251-255) that we do not recommend the LDSC-intercept estimates as the simple low-MAF method gives more consistent estimates. This improvement has been demonstrated in the bioRxiv preprint for experts in the field to evaluate.

2.5 Threshold for Θ

Q5a: What would be a threshold for Theta, above which one should be interested in following up on a SNP? I agree with the authors that it is very likely that any random region of the genome affects some complex trait. So, testing if Theta is greater than 0 seems to be a less interesting target. However, testing if Theta is greater than 0 does provide a baseline. The authors mentioned that they would emphasize the magnitude of Theta. However, they did not provide a threshold for Theta.

A: We agree with the reviewer that testing of Θ against zero provides information, while such information can be driven by the estimation uncertainty of $\hat{\Theta}$. In the TGCA setting, we care less about whether a locus is a true or false positive for the group of phenotypes, but rather which loci contribute the most to the phenotypes. When reporting the top loci, we use a threshold of $\hat{\Theta} > 2$ (Supplementary Table 2-4), which corresponds to the 99.97% quantile of the empirical distribution of all the estimated $\hat{\Theta}$ across the five trait domains. When integrating with tissue-specific cis-eQTL signals, many SNPs with $\hat{\Theta} < 2$ contributed to the analysis. So a hard “significance threshold” does not directly apply to Θ estimates. We also discuss (lines 244-250) that there is always a sensitivity-specificity trade-off when reporting SNPs or loci, so that the threshold is in principle arbitrary depending on the aim of the analysis.

2.6 Visualisation of simulation results

Q6: I appreciate the authors for perming the additional simulations. My only comment is to add a horizontal line at $y = 0$.

A: We thank the reviewer for this suggestion. Horizontal lines at $y = 0$ are added in the revised Figure 2 and Extended Data Figures 1-2, 17-18.

2.7 On the S-LDSC analysis

Q7: In the authors' response to my concerns regarding S-LDSC analysis, the authors satisfactorily addressed my first concern regarding the distribution of TGCA Z-scores. However, the author did not seem to address my 2nd concern of using GWAS summary statistics twice, once for creating the annotations, and once for enrichment analysis.

A: For this further question, we apologise that we did not to make it clear enough in the first response letter. In short, we removed the part of analysis where the reviewer concerned about using GWAS summary statistics twice in S-LDSC analysis. In fact, we did not "double-use" the GWAS summary statistics as we used UK Biobank GWAS summary statistics for TGCA and SNP annotations but non-UK-Biobank consortia GWAS summary statistics for S-LDSC heritability enrichment analysis. However, we found that part of analysis did not add much new knowledge compared to similar analysis done in the HOPS work by Reviewer #3, and this paper already had sufficient results, so we decided to remove those results.

3 Responses to Reviewer #3

In general, I am very satisfied with the authors' response to my comments. I particularly want to thank the authors for their positive and friendly responses to my request that my work be cited and to my signing my review. I strongly agree that this kind of exchange is good for the peer review process and for science generally. On a more personal level, in the past I have typically only signed my reviews when my identity is relevant to my comments (as it was here) and it often makes me nervous to do so. Getting a specific positive response to signing my review will make me more willing to sign my reviews in the future, and it is greatly appreciated.

The manuscript is substantially stronger and easier to read now as a result of the responses to mine and the other reviewers' comments. It is much clearer which portions are novel to this work and what the rationale is for the methodological choices the authors have made, and this also makes the sections of the manuscript describing applications of the new method much more compelling. I have the following minor comments as follow-ups to my original comments.

3.1 Defining "true Θ "

Q1: I am completely satisfied with the responses to the issues of model choice and robustness. The expanded simulations in particular are extremely compelling, and like the authors I am pleasantly surprised to see that the three-Gaussian model does seem to have general power to fit a large range of alternative models. My only real comment on these simulations is that the text should be clearer about how the "true Θ " is defined in these alternative models, and a brief comment on whether those Θ s are really equivalent across models (though I think it's probably correct that they are). The one thing I would still like to see simulations of is more heterogeneous values of the π parameters. However, the simulations now do cover a very wide range of parameters, and the three-Gaussian model does have some heterogeneity in the π parameters, so I don't think adding additional simulations is necessary for publication.

A: We thank the referee for appreciating our efforts in the first revision. We agree that we should define "true Θ " clearly for the alternative models. In this revision, we provide details of the definition of "true Θ " for each model considered, and we explain that the true values are comparable across different true models – Θ always measures the total genetic contribution as an average *non-null* genetic effect (lines 354-360 & 365-370).

3.2 Plain definition of Θ

Q2: I think the interpretation of Θ is a little clearer, and reading through the revised manuscript I no longer get the impression that it is somehow supposed to measure level of pleiotropy. I still think the idea of a "total genetic contribution" can be explained a little more clearly, to make it clear to readers who can't follow the math that the goal is to get a kind of combined GWAS association score that measures the aggregate effect of a SNP across multiple traits. This probably only requires one additional sentence in the introduction or conclusion repeating this in plain language.

A: We agree with the referee that we indeed seem to miss one plain sentence in Introduction. In this revision, we add an explanation of the total genetic contribution idea in the last paragraph of Introduction (lines 47-48). The concept is emphasised again in the first paragraph of Discussion.

3.3 P-values for the null hypothesis $\Theta = 0$

Q3: I am satisfied with the response to my comment about the definition of p-values (that $\Theta = 0$ is an interesting null hypothesis to test but that ultimately the Θ score itself is more important than the p-value) but that response is only reflected in a few of the many places in the revised manuscript where p-values are mentioned. I won't suggest that the p-values be removed from the manuscript entirely, but it might be helpful to change the way they are referenced, so that they are referred to throughout the manuscript as p-values for the null hypothesis of $\Theta = 0$, rather than as p-values for the Θ statistic. The most egregious use of these p-values is in Figure 4,

where the p-values are used to describe the distribution of Θ scores and the Θ scores themselves are not shown. If possible, that figure and that analysis should be reworked so that the distributions are displayed as Θ values rather than p-values.

A: We thank the referee for this comment. We have now tried to state clearly throughout the paper that the p-value corresponds to a null hypothesis of $\Theta = 0$. To not overuse the p-values, we remade Figure 4 using the squared Z-scores (χ^2 's) for $\hat{\Theta}$, where the Z-scores are essentially the Θ estimates scaled by their estimated standard errors. We would like to note here that for this analysis – the enrichment of TGCA signals on cis-eQTL of tissue-specifically expressed genes – many variants with not so high Θ values are evaluated. Thus it is necessary to consider the estimation uncertainty, measured by the standard errors, otherwise the estimates themselves would be too noisy for a visualisation purpose.

3.4 Final remarks

Overall, again, I think the manuscript is very much improved and I'm very impressed with the work the authors have put into addressing my comments and the comments of the other reviewers. I look forward to seeing the final version and to seeing future work on this subject.

Daniel M. Jordan

A: We thank the referee for being so appreciated by our hard work and these encouraging words. The field of complex traits genetics is moving into an exciting, although complicated, era. With the fast developing big data resource, we believe future work on this topic of shared genetic architecture across phenotypes would reveal more and more new knowledge for understanding the complex biology.

Dr. Xia Shen, on behalf of the co-authors

Reviewer #2 (Remarks to the Author):

The authors have adequately addressed my concerns. And I am happy to recommend acceptance. I also appreciate the authors for highlighting their changes.

Reviewer #3 (Remarks to the Author):

I am satisfied with the authors' response to my comments, and I do not have any additional comments. Thank you for all the hard work you've put into addressing these comments.

Daniel M Jordan

Point-to-point responses to reviewers

for “Total genetic contribution assessment across the human genome” (Final revision)

March 15, 2021

Contents

1 Responses to Reviewer #2	1
2 Responses to Reviewer #3	1

1 Responses to Reviewer #2

The authors have adequately addressed my concerns. And I am happy to recommend acceptance. I also appreciate the authors for highlighting their changes.

We are glad that the reviewer is satisfied with our revision.

2 Responses to Reviewer #3

I am satisfied with the authors’ response to my comments, and I do not have any additional comments. Thank you for all the hard work you’ve put into addressing these comments.

Daniel M Jordan

We are glad that the reviewer is satisfied with our revision. Again, we thank the reviewer for signing the report, allowing for open scientific exchange.

Dr. Xia Shen, on behalf of the co-authors